



# Observations of microphysical properties and radiative effects of contrail cirrus and natural cirrus over the North Atlantic

Ziming Wang[1,2], Luca Bugliaro[1], Tina Jurkat-Witschas[1], Romy Heller[1], Ulrike Burkhardt[1], Helmut Ziereis[1], Georgios Dekoutsidis[1], Martin Wirth[1], Silke Groß[1], Simon Kirschler[1,3], Stefan Kaufmann[1], Christiane Voigt[1,3]

[1]Institute of Atmospheric Physics, German Aerospace Center (DLR), Oberpfaffenhofen, 82234, Germany
[2]Meteorological Institute Munich, Ludwig-Maximilians-Universität München, Munich, 80333, Germany
[2]Institute of Atmospheric Physics, Johannes Gutenberg University, Mainz, 55128, Germany

*Correspondence to*: Ziming Wang (Ziming.Wang@dlr.de) and Luca Bugliaro (Luca.Bugliaro@dlr.de)

**Abstract.** Contrail cirrus constitute the largest radiative forcing (RF) component of the aviation effect on climate. However, the difference of microphysical properties and radiative effects between contrails, contrail cirrus and natural cirrus clouds are still not completely resolved. Motivated by these uncertainties, we investigate the cirrus perturbed by aviation in the North Atlantic Region on 26 March 2014 during the Mid Latitude Cirrus (ML-CIRRUS) experiment. In the synoptic context of a ridge cirrus cloud, an extended thin ice cloud with many persistent contrails can be observed for many hours with the geostationary Meteosat Second Generation (MSG)/Spinning Enhanced Visible and InfraRed Imager (SEVIRI) from the morning hours until dissipation close to 14 UTC. Airborne lidar observations aboard the German High Altitude and LOng Range Research Aircraft (HALO) suggest that this cloud is mainly of anthropogenic origin. We develop a new method to distinguish between contrails, contrail cirrus and natural cirrus based on in situ measurements of ice number and NO gas concentrations. It turns out that effective radii ($R_{eff}$) of contrail cirrus and contrails are in the range of 3 to 53 µm and about 18% smaller than that of natural cirrus, hence a difference in $R_{eff}$ is still present. Ice particle sizes in contrail cirrus are on average 114% larger than in contrails. The optical thickness of natural cirrus, contrail cirrus and contrails derived from satellite data has similar distributions with average values of 0.21, 0.24 and 0.15 for these three cloud types, respectively. As for radiative effects, a new method to estimate top-of-atmosphere instantaneous RF in the solar and thermal range is developed based on radiative transfer model simulations exploiting in situ and lidar measurements, satellite observations and ERA5 reanalysis data for both cirrus and cirrus-free regions. Broadband irradiances estimated from our simulations compare well with satellite observations from MSG and the Geostationary Earth Radiation Budget (GERB), indicating that our method provides a good representation of the real atmosphere and can thus be used to determine RF of ice clouds probed during this flight. Contrails net RF is smaller by a factor of 4 compared to contrail cirrus. On average, the net RF of contrails and contrail cirrus is more strongly warming than that of natural cirrus. For a larger spatial area around the flight path, the RF is well related to that along the flight track. We find warming contrail cirrus and cirrus in the early morning and cooling contrail cirrus and cirrus during the day. The results will be valuable for research to constrain uncertainties in the assessment of climate impacts of natural cirrus and contrail cirrus and for the formulation and evaluation of contrail mitigation options.

## 1 Introduction

Aviation accounts for about 3.5% of global effective radiative forcing (ERF) from all human activities (Lee et al., 2021). Among the individual aviation contributions, contrail cirrus contributes to more than 50% of the total aviation ERF component (Lee et al., 2021). Contrail cirrus and natural cirrus are both high level clouds composed of ice crystals that form and evolve in ice supersaturated regions (ISSRs) (Minnis et al., 2004). Contrails form when the hot and humid jet engine exhaust at cruise levels mixes with the cool ambient atmosphere, which at temperatures lower than the Schmidt-Appleman criterion (Schumann, 1996) can lead to a local liquid saturation of the plume. The emitted soot particles act as condensation nuclei to form liquid droplets that freeze subsequently in the young contrails (Bier et al., 2017; Kärcher, 2018; Kärcher and Voigt., 2018; Kleine et



al., 2018). In ISSRs, persistent contrails may grow and spread out to form contrail-induced cirrus (Burkhardt and Kärcher, 2011; Schumann et al., 2017). By reflecting incident solar radiation and trapping upwelling radiation within the Earth's atmosphere, they result in an imbalance of radiation budget in both the shortwave (SW) and longwave (LW) solar spectrum (Stuber et al., 2006). The net radiative forcing (RF) is positive (Rädel and Shine, 2008; Burkhardt et al., 2018; Gettelman et
al., 2021) and RF due to contrail cirrus greatly exceeds that from linear contrails (Burkhardt and Kärcher, 2011; Voigt et al., 2011; Burkhardt et al., 2018). Due to various reasons, feedback of natural clouds, the radiative response to the presence of contrail cirrus, the uncertainty in upper tropospheric water budget (including initial contrail properties, contrail cirrus properties and relative humidity), contrail cirrus schemes (see Lee et al., 2021), and the challenges in measuring and separating contrail cirrus from natural cirrus, a best central estimate of the contrail cirrus RF remains challenging, further limiting
projections of aviation climate impact and formulations of mitigation options other than carbon dioxide ($CO_2$) emissions (Voigt et al., 2021). Knowledge gaps still exist regarding the large variability in the contrail life cycles (Bier et al., 2017), and optical properties which then determine their radiative response to the climate system (Forster et al., 2007; Grewe et al., 2017).

Contrails may form in, overlap, merge, and interact with natural cirrus (Duda et al., 2001; Vázquez-Navarro et al., 2015). Contrail cirrus primarily differs from natural cirrus by their larger ice number concentrations (N) (Heymsfield et al., 2010;
Voigt et al., 2010; Voigt et al., 2017; Sanz-Morère et al., 2020). Consequently, microphysical process rates, which control their life cycle, and radiative effects can be very different to those in natural cirrus and are dependent on soot number emissions (Bier et al., 2017). Besides soot particles emitted from aviation, volatile aerosols inject into the upper troposphere (UT) in ice subsaturated conditions and can alter cirrus clouds and liquid water clouds at a later stage (Hendricks et al., 2005; Lee et al., 2009; Schumann et al. 2015; Urbanek et al., 2018; Righi et al., 2021). Furthermore, natural cirrus locally optically thickens by
embedded contrails (Tesche et al., 2016; Quaas et al., 2021; Schumann et al., 2021a, b), but contrails can also cause a decrease in natural cloudiness (Burkhardt and Kärcher, 2011). These aspects are still subject of current research (e.g., Verma and Burkhardt, 2022).

Aircraft and spaceborne measurements have provided detailed properties of contrail cirrus. Firstly, contrail cirrus can be detected and separated from natural cirrus to some extent in in situ measurements by combining ice crystal microphysical data
with observations of aircraft emissions such as nitrogen oxides ($NO_x$) or aerosols (Voigt et al., 2017; Voigt et al., 2021; Bräuer et al., 2021a). Schumann et al. (2017) and Heymsfield et al (2010) provide comprehensive overviews of contrail and contrail cirrus properties and extensive data sets exist on their microphysical properties (e.g. Petzold et al., 1997; Baumgardner and Gandrud, 1998; Jensen et al., 1998a, b; Voigt et al., 2010; Bräuer et al., 2021b), their particle shapes (Gayet et al., 2012; Järvinen et al., 2016; Sanz-Morère et al., 2020) and optical properties (Chauvigné et al., 2018), as well as the aviation influence
on them (Jeßberger et al., 2013; Schumann and Graf, 2013; Marjani et al., 2022). Recent attempts used the reduced air traffic situation due to the COVID-19 pandemic to evaluate the aircraft impact on cirrus and climate (Gettelman et al., 2021; Li and Groß, 2021; Quaas et al., 2021; Schumann et al., 2021b; Meijer et al., 2022; Voigt et al., 2022). While the aircraft impact on clouds is confirmed by those studies, the magnitude of the reduced contrail cirrus forcing is variable and depends on the region, season and the method used to derive the impact. Some studies have taken North Atlantic and North America with the largest
air traffic density as target regions, and analysed the diurnal cycle of contrail cirrus coverage, outgoing radiation, and properties during several contrail outbreaks (Duda et al., 2004; Atlas et al., 2006; Haywood et al., 2009; Graf and Schumann, 2012; Duda et al., 2013; Minnis et al., 2013; Schumann and Graf, 2013).

Early climate models estimated contrail cirrus RF through associating air traffic with regional cirrus coverage and assumed equal radiative efficiencies of contrails and contrail cirrus (Stordal et al., 2005; Rädel and Shine, 2008). Later, the global
climate models represented contrail cirrus as a separate cloud class (Burkhardt and Kärcher, 2011; Bock and Burkhardt, 2016). In another study, contrail cirrus is simply treated as a source for the ice crystal budget of the natural cirrus, mixing the microphysical properties of natural ice clouds and contrail cirrus (Chen et al., 2012). Hence, despite substantial progress in recent years, the characterization of geometrical, optical, and microphysical properties of contrails and their evolution within



natural cirrus fields as well as the calculation of the radiative impact are still subject to large uncertainties due to instrumental and model limitations and the large number of variables influencing the contrail life cycle (Voigt et al., 2017; Chauvigné et al., 2018; Kaufmann et al., 2018; Rodríguez De León et al., 2018; Gierens et al., 2020).

In this study, we use in situ data measured during the Mid Latitude Cirrus (ML-CIRRUS) experiment (Voigt et al., 2017) from the German High Altitude and LOng Range Research Aircraft (HALO) and simultaneous remote sensing observations (Bugliaro et al., 2011; Vázquez-Navarro et al. 2013; Strandgren et al., 2017) with high temporal resolution from the geostationary Meteosat Second Generation (MSG). In particular, we concentrate on one flight on 26 March 2014 over the North Atlantic Region (NAR) just off the coast of Ireland where most of the air traffic from Europe to the US and viceversa takes place. This situation enables us to investigate properties and radiative effects of contrails, contrail cirrus and ambient natural cirrus. In particular, we develop a new method to classify ice crystals along the HALO flight track from in situ measurements based on enhanced NO aircraft gas emissions and ice number concentrations N into three classes that are representative for (1) contrails, (2) contrail cirrus and (3) natural cirrus where the effect of aircraft emissions is not directly observable. For these classes we evaluate microphysical ice crystal properties (effective radius, $R_{eff}$) and relate them to relative humidity over ice (RHi). From satellite remote sensing we evaluate ice optical thickness (IOT), $R_{eff}$ of the ice crystals, reflected solar radiation (RSR) and outgoing longwave radiation (OLR) along the flight path. Furthermore, we also develop a new approach for the determination of ice cloud RF that combines in situ and satellite observations with a radiative transfer model (RTM). To this end, we use reanalysis data from ERA5 (Hersbach et al. 2020). Since ERA5 does not simulate the effect of air traffic on clouds, we collect atmospheric profiles of water vapour, liquid and ice clouds from this reanalysis and combine them with the in situ and spaceborne observations to provide inputs to the RTM to compute RSR and OLR. After checking the consistency of our radiative transfer calculations with the corresponding observations, we are in the position to compute instantaneous RF in the SW and LW spectral range along the flight path of HALO by excluding the ice cloud layer (contrails, contrail cirrus and natural cirrus) at the flight level from the radiative transfer calculations, thus yielding a consistent ice-cloud-free irradiance. In a second step, we extend these calculations to an area encompassing the HALO flight path to compute the diurnal cycle of RF in that region.

Detailed information about airborne and satellite datasets, as well as cirrus remote sensing techniques and the RTM are presented in section 2. Microphysical properties of contrail cirrus and natural cirrus, consisting of cirrus classification, and further differences between contrail cirrus and natural cirrus, are summarized in section 3. Radiative effects of contrail cirrus and natural cirrus, including the top-of-atmosphere (TOA) radiation estimation method, detailed RTM calculations of RF along HALO flight, and the investigation of cirrus spatial pattern are organized in section 4. Finally, summary and conclusions are provided in section 5.

## 2 Data and approaches

### 2.1 Airborne measurements

During ML-CIRRUS, the German research aircraft HALO was equipped with a comprehensive suite of novel particle measurement sondes, and obtained a broad dataset of microphysical properties of natural cirrus and contrail cirrus for process studies and climatological analyses. Ice number concentrations N, $R_{eff}$, size distributions, ice or liquid water content (IWC and LWC) and extinction are derived from measurements of CAS-DPOL (Cloud and Aerosol Spectrometer with Detector for Polarization) for particles from 3 to 50 µm and CIP (Cloud Imaging Probe) for the size range of 15 and 960 µm (diameter as the maximum dimension). CAS-DPOL measures the forward scattered light of particles when they pass through a laser beam (Baumgardner et al., 2011). The uncertainty of the particle size measurements is ±16% (Kleine et al., 2018). Using 64-element linear photodiode arrays, the CIP acquires two-dimensional shadow images of particles (De Reus et al., 2009). The size resolution is 15 µm with the uncertainty decreasing considerably with diameter, reaching ±15 µm when particles are larger



than 50 µm. N from CAS-DPOL, denoted by $N_{CAS}$ in the following, and from CIP ($N_{CIP}$) are also combined to an overall N that considers the particle size overlap of the two instruments. The aspherical fraction of particles with $R_{eff}$ larger than 3 µm measured by CAS-DPOL is determined by the ratio between perpendicularly polarized light and the forward scattering light. Based on measurements in the Aerosol Interaction and Dynamics in the Atmosphere (AIDA) cloud chamber described in Järvinen et al. (2016), particles with a polarization ratio larger than the 1σ range of size-dependent thresholds are categorized

as aspherical. Validation of the measurements, particular for the smaller spherical particles, have been performed taking atmospheric and cloud chamber measurements into account (Braga et al., 2017a, b).

As for ambient conditions, the AIMS (Atmospheric Ionization Mass Spectrometer, Jurkat et al., 2016, Kaufmann et al., 2016) was applied to measure the actual water vapor concentration from ambient air using a backward heated inlet. The range of detection is between 1 and 500 ppm with an overall accuracy from 7% to 10%. Static pressure and temperature, measured by

the BAHAMAS (Basis HALO Measurement and Sensor System, Krautstrunk and Giez, 2012) with an accuracy of 0.3 hPa and 0.5 K, were used to convert water vapor concentration to RHi. NO was measured by the conventional chemiluminescence technique using AENEAS (AtmosphEric Nitrogen oxides mEAsuring System chemiluminescence detector, Ziereis et al., 2000) as the aircraft tracer. The upper limit of the measuring range limit is 60 ppbv. The overall uncertainty is 8% for NO for concentration levels of 0.5 ppbv (Stratmann et al., 2006).

Backscatter profiles of clouds and aerosol were acquired by the lidar system WALES (Water vapor Lidar Experiment in Space, Wirth et al., 2009) at the wavelengths of 532 (Esselborn et al., 2008) and 1064 nm. In this study backscatter is used to extract information about the cirrus cloud structures, such as cloud top height (CTH), geometrical depth and others. Apart from the backscatter coefficient, WALES also provides 2D measurements of the water vapor mixing ratio and aerosol particle linear depolarization ratio. The backscatter ratio and aerosol depolarization are used to create a cloud mask, which helps keep only

ice clouds. For these clouds the RHi is calculated from the measured water vapor mixing ratio and collocated model temperatures from the European Centre for Medium-Range Weather Forecasts (ECMWF). This instrument and method have also been applied by Groß et al. (2014) and Urbanek et al. (2018), who found that the lidar measurements were accurate when compared with in situ data. The statistical error in the retrieval of water vapor by WALES is estimated to be about 5% (Kiemle et al., 2008) and the ECMWF temperatures induce an error of around 10-15% in the final RHi values (Groß et al., 2014).

**2.2 Satellite remote sensing**

The Spinning Enhanced Visible and InfraRed Imager (SEVIRI) is the primary instrument aboard the geostationary MSG satellites, which provides observations of the Earth disk every 15 min from 3 solar and 8 thermal channels with 3 km sampling distance at nadir, and one High Resolution Visible channel with 1 km spatial resolution (Schmetz et al., 2002). We use MSG-3 / Meteosat-10 observations for the study on 26 March 2014 with a temporal resolution of 15 min.

**2.2.1 CiPS**

CiPS (Cirrus Properties from SEVIRI) detects cirrus with their transparency information and retrieves the corresponding CTH, IOT, ice $R_{eff}$ and ice water path (Strandgren et al., 2017). It consists of four artificial neural networks trained using SEVIRI thermal observations, CALIPSO (Cloud-Aerosol Lidar and Infrared Pathfinder Satellite Observations) cloud products, and ECMWF ERA5 surface temperature and auxiliary data. CiPS has been especially developed for thin cirrus and validated

against CALIPSO. CiPS detects 20%, 70% and 85% of the ice clouds with an IOT of 0.01, 0.1 and 0.2 respectively. For IOT between 0.35 and 1.8 CiPS has a mean absolute deviation smaller than 50%. This value increases for IOT between 0.07 and 0.35. For CTHs larger than approx. 8 km, CTH has an absolute percentage error of 10%, with underestimation for CTH > 10 km at 50° N and overestimation for CTH < 10 km at the same latitude. An example is shown in Fig. 1.



### 2.2.2 GERB and RRUMS

The GERB (Geostationary Earth Radiation Budget) instrument measures broadband solar and thermal components which are subsequently converted to outgoing and reflected fluxes considering the cloud properties and surface type detected by SEVIRI (Harries et al., 2005). GERB's sampling distance is larger than that of SEVIRI with a spatial resolution of 44.6 km × 39.3 km but the same image repeats the cycle of 15 min. During the processing, the finer spatial resolution of the SEVIRI data is used to improve the original GERB resolution and results in GERB products for 3x3 SEVIRI pixels. In general, the GERB SW and

LW fluxes are found to be 7.5% higher and 1.3% lower respectively, compared to products from the Clouds and the Earth's Radiant Energy System (CERES), whose data records are from polar orbiting satellites (Wielicki et al., 1996). The bias of CERES is estimated to 1% and 0.5% for OLR and RSR, respectively.

   Based on a linear regression and a neural network, an algorithm named RRUMS (Rapid Retrieval of Upwelling Irradiances from MSG/SEVIRI) was also developed, which estimates OLR and RSR at TOA from SEVIRI at pixel levels. RRUMS shows

excellent agreement with OLR from CERES within 1% and a systematic overestimation of RSR from CERES or GERB of 5% to 10% in the worst cases under high viewing angles (Vázquez-Navarro et al., 2013).

### 2.2.3 Collocation

   We exploited ground positions of ice clouds from ML-CIRRUS measurements and CTHs estimated from WALES observations to collocate the clouds probed by HALO with the SEVIRI observations. Thanks to the 15 min repeat cycle of the

geostationary imager, temporal shifts up to 7.5 min result from this procedure. Since the temporal frequency of in situ observations is 1 Hz, various HALO values are located inside each SEVIRI pixel. Of course, HALO measurements do not fill SEVIRI pixels such that they are representative only of a part of them. In general, statistics or time series of observations are produced with the original temporal resolution of the given instrument. However, when one in situ IOT or $R_{eff}$ value for a SEVIRI pixel is needed (see e.g. Sect. 3.2) then averages are computed.

### 2.3 Radiative transfer model

   To calculate broadband solar and thermal irradiances at TOA for ice particles probed during the HALO flight, a sophisticated radiative transfer package libRadtran is used (Mayer and Kylling, 2005; Emde et al., 2016). Water and ice clouds are represented in this model detailly and realistically. Optical properties of water droplets are computed using the Mie theory and tabulated as a function of wavelength and $R_{eff}$. Ice crystals are not spherical in shape and habits (Letu et al., 2016), and for

this simulation the parameterisation of Baum et al. (2011) for ice crystal habits has been employed to define the conversion from optical to microphysical properties. In analogy to the MODIS products (Yang et al., 2018), we select rough aggregates for ice crystal shape (see Sect. 4.2 for a discussion about this choice). The selected one-dimensional radiative transfer solver is DISORT (Discrete Ordinate Radiative Transfer) 2.0 by Stamnes et al. (2000) with 16 streams. And libRadtran recommends the REPTRAN band parameterization with a spectral resolution of 15 cm$^{-1}$ for spectral calculations (Buehler et al., 2010;

Gasteiger et al., 2014).

### 3 Microphysical properties of contrail cirrus and natural cirrus

   In this section we describe how airborne data from HALO and satellite remote sensing outputs from CiPS are combined to understand microphysical properties of contrail cirrus and natural cirrus, and the corresponding variation when contrails transit to contrail cirrus within the contrail life cycle.

### 3.1 General situation

In the night and the very early hours of 26 March 2014 a ridge cirrus cloud band built up North of Ireland down to the Southern tip of Portugal (Fig. 1). North of Ireland close to Iceland as well as to the South-West of Ireland this cirrus cloud is thicker (optical thickness larger than 1 at 10:45 UTC, Fig. 1c) while exactly off the coast of Ireland, in coincidence with the eastbound morning and westbound afternoon air traffic to and from the US, many linear structures can be seen in SEVIRI observations (Fig.1 is enlarged in Fig.2) and the ice cloud is thinner (optical thickness around 0.3 at 10:45 UTC). Considering that the peak of eastbound morning air traffic is approx. at 3 UTC (Graf and Schumann, 2012), these contrails might be those identified in this study, with maybe some older contrails from the westbound traffic in the afternoon of 25 March, which however peaks already at 13 UTC. This cloud band evolves with time towards the South, and in correspondence of the Ireland coast the ice clouds dissipate until approx. 14 UTC. In addition, this thin bluish/violet ice cloud band partly overlaps with a liquid water cloud field below (yellowish clouds in Fig. 1a). At the same time, the weather condition over Europe is characterized by a cyclone affecting the Mediterranean Sea with most of mainland Europe covered by liquid clouds and cirrus over the Eastern Atlantic Ocean.

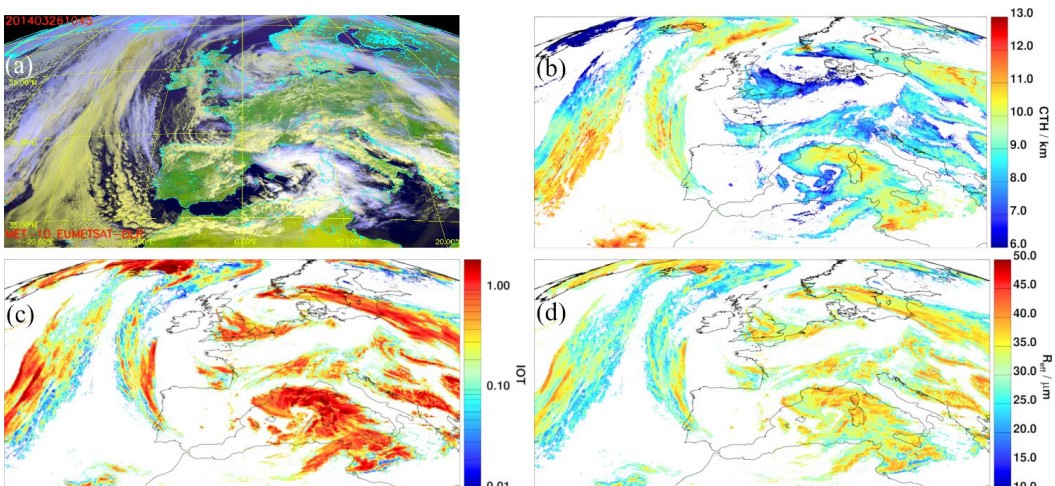

**Figure 1: (a) The false color RGB image from MSG/SEVIRI on 26 March 2014 at 10:45 UTC showing Europe and the Eastern part of the North Atlantic Ocean, (b) corresponding ice clouds CTH, (c) IOT and (d) $R_{eff}$ from CiPS**

Due to its moderate spatial resolution (Sect. 2.2), SEVIRI can only observe contrails that have already grown larger (and thicker) while young contrails are usually missed (Mannstein et al. 2010). Thus, the presence of these numerous persistent contrails as well as the simultaneous evolution of the cloud band indicate that contrails and contrail cirrus / persistent contrails maybe embedded in or on top of / below cirrus particles of natural origin. The relative vertical location of contrails and natural clouds is difficult to determine from the passive remote sensing perspective as well as the origin of ice particles, unless the typical linear shape of contrails can be directly observed. Nevertheless, the satellite observations indicate that because of these favourable meteorological conditions and the relatively high air traffic density in this region, contrails from various aircrafts could form over a long time period (~12 h) that might overlap in the vertical direction or that are formed inside / on top of / below "the remnants" of previous contrails. Thus, air traffic in this area could have a strong impact on cloudiness.

On 26 March 2014 the HALO aircraft started from Oberpfaffenhofen in Germany for the first flight of the ML-CIRRUS campaign (Voigt et al., 2017) at approximately 05:30 UTC and probed the cirrus over NAR from around 08:00 to 11:30 UTC with a race track pattern between approx. 51.5°N and 54°N at a longitude of ca. -14°E (-13.6 to -14.4°E), see also Voigt at al. (2017, Fig.4). In this area, HALO flew 3 in situ legs split by 3 lidar legs.

Figure 2 presents the temporal variation of contrails and surrounding clouds with the real-time HALO flight path from 08:00 (HALO entering the NAR) to 12:00 UTC (HALO leaving the NAR). For the sake of presentation, we show hourly plots, but



for the analyses exposed in the paper we use the full 15 min resolution. To each SEVIRI observation we assign the closest HALO measurements (Sect. 2.2.3) that thus differ in acquisition time from the SEVIRI images by at most 7.5 min. The HALO path already flown - i.e. older that the acquisition time of SEVIRI minus 7.5 min - is plotted in red, while the HALO flight track corresponding to the current satellite images – i.e. HALO path in the time range between SEVIRI acquisition time plus/minus 7.5 min - is plotted as blue. Notice also the nominal slot time of SEVIRI of e.g. 09:00 UTC corresponds to a real acquisition time of approx. 09:12 UTC. In addition, we have drawn a red rectangle around the flight path that is investigated in Sect. 4.4 in more detail but that serves here as orientation to easily capture the temporal evolution of the ice clouds. In the brightness temperature difference (BTD) images – where SEVIRI brightness temperatures at 10.8 µm and 12.0 µm are subtracted from each other - the black areas are caused by the low-level clouds, while the bright pixels correspond to thin cirrus. Small ice crystals, for instance in contrails, correspond to the largest BTDs. Around 08:00 UTC, many thin lines (contrails) are seen in the BTD picture that are intersected perpendicularly by the HALO route (top row). However, these contrails form a thin cirrus layer, with anthropogenic but maybe also with naturally nucleated ice crystals (see Fig.3). In the South, the ridge cirrus is thicker and no contrails can be observed. At 09:00 and 10:00 UTC, the flight area is dominated by contrails. With time, from 08:00 to 12:00 UTC the typical contrail lines become always fainter and less in number. Thus, contrails begin either to dissipate (see low IOT) or to lose completely their linear shape due to wind shear such that they turn out to be undistinguishable in the red area at 12:00 UTC (and only in part at 11:00 UTC). However, contrails can be still observed in the north-eastern part of the satellite images (outside of the red areas) even at later times.

**Figure 2: Temporal variation of contrail cirrus and surrounding clouds from MSG/SEVIRI observations over the NAR corridor on 26 March 2014. Left: RGB-composite with overlaid cirrus and low-level liquid clouds pixels. Middle: 10.8 µm and 12.0 µm BTD (K) with overlaid cirrus pixels. Right: IOT from CiPS. Top to bottom: 08:00, 09:00, 10:00, 11:00 and 12:00 UTC. The red box is the area investigated in Sect. 4.4. The red-blue line represents the HALO flight path. The HALO positions around the SEVIRI acquisition time plus/minus 7.5 min are plotted as a blue line, and the older HALO positions are plotted as a red line. The cross indicates the time of simultaneous SEVIRI and HALO measurements.**



**Figure 3: The three panels show the three lidar legs with backscatter ratios at 1064 nm and 532 nm. Leg 2 is much shorter than the other two legs, and leg 1 starts before 8:30 UTC, which is the start time of the evaluation presented in the paper.**

Figure 3 presents the backscatter ratio at 532 and 1064 nm from the WALES lidar measurements in that sequences. The lidar data show that the geometrical thickness of high-level clouds reduced from ~2.0 km at 08:30 UTC to ~1.5 km at around 11:00 UTC. The temporal evolution of CTH from WALES shows that the cloud firstly reached up to approximately 12 km and



slowly descended to slightly above 11.5 km, with backscatter values becoming smaller with time, in line with the passive
observations that indicate dissipation of the cloud during the day. Mid-level clouds appeared occasionally at a height between
4 and 6 km (not shown). In the following we concentrate on the time period between 08:30 and 11:30 UTC.

Considering the three WALES legs at 1064 nm and 532 nm in more detail (Fig. 3), one can observe in leg 1 between ca. 08:10
and 08:25 UTC a series of spots with high backscatter at a height of approx. 11 km below a thin ice cloud layer top at 12 km
and with connected fall streaks that extend further down for up to 1 km. These structures resemble those in the large eddy
simulations by Unterstrasser et al. (2017a, b) where they considered contrail formation followed by homogenous nucleation
of a natural cirrus cloud. This might be the case here as well: contrails formed first and reduced RHi and then the cirrus formed
around it. However, contrails are so many and so close to each other that undisturbed ice clouds do not seem to be visible.
Thus, these bright spots with their fall streaks can be very probably identified as contrails, with many small ice particles causing
high backscatter and with larger ice crystals sedimenting down. For leg 1, these contrails seem to make up the majority of the
cloud in this temporal interval, with maybe homogeneous nucleation on top, although some isolated spots are observable in
the top ice layer as well. The presence of developed fall streaks is a hint that these contrails are not very young (> 30 min). In
addition, not only in this time frame but also afterwards during leg 1 some single spots with elevated backscatter ratios can be
observed. Unfortunately, the high WALES backscatter inhibits the determination of RHi (Fig. 4) for these bright spots, but the
upper level of the clouds, just below the potential contrails, show frequent ice supersaturation, with RHi above 100%. In the
lower cloud part, where many ice crystals are still encountered, RHi is below saturation, in large parts also below 90%.

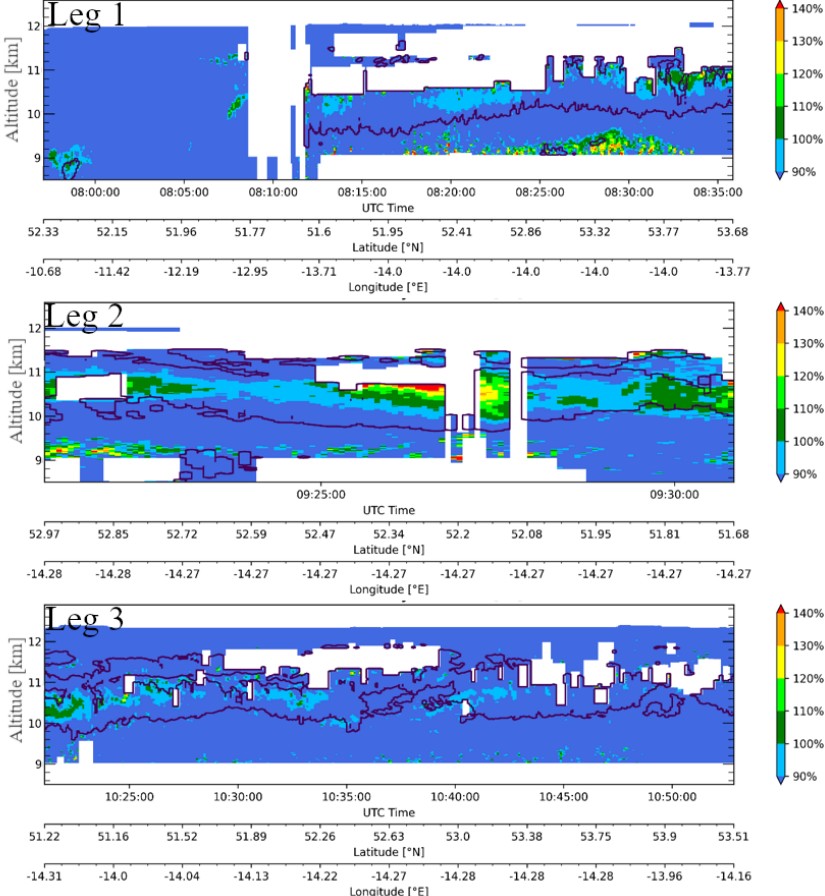

**Figure 4: The three panels show the RHi from Fig. 3 in more detail. The black contour represents the cloud edge. White areas are
due to lack of measurements in these regions, because of detector saturation.**





Lidar leg 2, much shorter than the other two, also shows the presence of some spots with elevated backscatter ratios, although
less numerous, at an altitude of ca. 11 km at around 09:26 UTC. Unlike leg 1, many high backscatter spots do not show
extended fall streaks, suggesting the presence of younger contrails in addition to older ones. This is supported by the fact that
corresponding RHi in Fig. 4 seems to reach very high levels (RHi > 140%), which makes natural nucleation events possible.
Furthermore, the top ice layer above ca. 11.5 km also contains some very high backscatter areas. Finally, also above 9 km
some clouds are visible that resemble contrails (or a dissipating cirrus). Thus, in this leg the contrail-like structures do not
seem to fill the entire cloud as in leg 1, but there are indications of young and older contrails, maybe in addition to natural
cirrus. The last lidar leg - leg 3 - took place approx. 1 h later and also shows various bright spots at different levels, from 10 to
almost 12 km, and elongated vertical structures that remind of those in leg 1. The cloud as a whole is slightly lower that in leg
1 with ragged edges especially at its lower border, suggesting that the cloud is sublimating and thinning out. This is confirmed
by the RHi observations of WALES for leg 3, where subsaturation is indicated especially at the lower and upper edges, while
at altitudes around 10.5 km at selected locations RHi reaches saturation, at least there where RHi could be determined (in Fig.
2 CiPS also shows a decreasing cirrus cloud cover with time). In the upper part of the cloud, where RHi could not be determined,
some bright spots are also observed. Overall, during all three lidar legs RHi is below saturation in large parts of the cloud.
Finally, it is interesting to note that many of the bright spots appear at latitudes around 52°N corresponding to the southern
part of the race track flown by HALO. Of course, one has to consider that these 3 lidar legs were not flown at the same position
and that the cloud evolved in time due to microphysical processes and dynamics. Therefore, these WALES measurements are
representative for the cloud probed between 08:00 and 11:30 UTC but can neither be directly intercompared nor directly
compared to in situ observations taken in between.

### 3.2 Properties of natural cirrus, contrails and contrail cirrus

In this region of sustained air traffic, with aircraft flying westbound to the US in the afternoon and eastbound to Europe in the
morning, new contrails might form in the ice cloud remnants of previous contrail cirrus generated by earlier aircrafts. We use
in situ measurements to distinguish between contrails, contrail cirrus and natural cirrus at the flight level of HALO. Notice
however that we cannot unambiguously attribute cirrus to "natural cirrus" since both old contrail cirrus and nucleation on
natural or anthropogenic aerosols transported from other sources might have taken place. At first, we consider in Fig. 5a all in
situ measurements to obtain a general overview of $R_{eff}$ against N from the combination of CAS and CIP about the cloud
properties collected during the three legs. In this combined dataset, one can notice first that both $R_{eff}$ and especially N are not
particularly high. While $R_{eff}$ up to 60 µm are encountered in natural cirrus clouds and low $R_{eff}$<5 µm are typical for contrails,
N in contrails can grow much larger than 10-20 cm$^{-3}$ as seen here (see discussion in Sect. 3.2.1). Furthermore, it is evident that
$R_{eff}$ lower than approx. 30 µm cannot be effectively detected by CIP while the CAS instrument is not able to identify ice
number concentrations N lower than 0.01 cm$^{-3}$. As a result, there is a range of small N and small $R_{eff}$ that cannot be observed
in this study (indicated in Fig. 5a as "No data"). Nevertheless, in general Fig. 5a shows that low concentrations are associated
with $R_{eff}$ up to 60 µm, and most values concentrate around 45 µm, with increasing concentrations $R_{eff}$ becoming smaller and
falling below 10 µm. Only few observations correspond to high N and low $R_{eff}$ and they are expected to represent contrails.
Thus, it is a quite homogeneous distribution of $R_{eff}$, with most occurrences between 20 and 45 µm, but also with a smaller peak
for $R_{eff}$ < 10 µm. N extends over various orders of magnitude, with most occurrences between 0.01 and 0.5 cm$^{-3}$. Single vertical
stripes for N close to 0.02 cm$^{-3}$ are due to the CAS resolution (sampling volume).





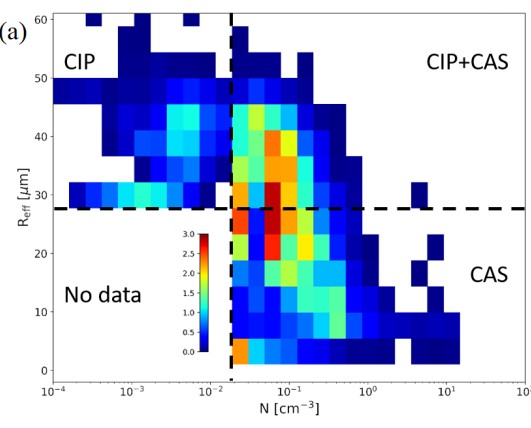

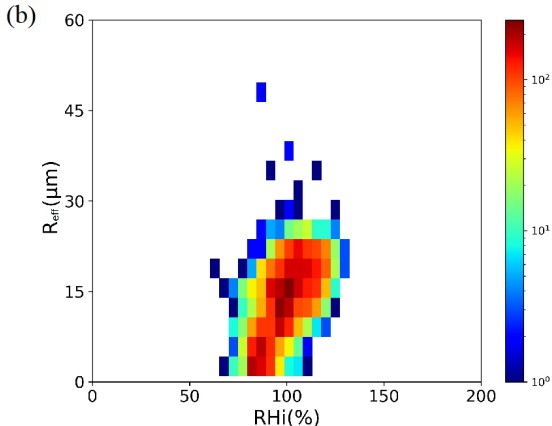

**Figure 5: (a) Combined occurrences for $R_{eff}$ and their corresponding N (particle concentrations from the cloud probes CAS and CIP) for all 1-sec data between 08:30 and 11:30 UTC. (b) $R_{eff}$ occurrence distribution versus RHi.**

Figure 5b also shows that over the entire flight path $R_{eff}$ increases with RHi, as ice supersaturation supplies the water vapor for the growth of ice crystals, while subsaturated conditions lead to sublimation and evaporation. Nevertheless, still many ice crystals are observed in subsaturated conditions.

### 3.2.1 Identification of natural cirrus, contrails and contrail cirrus

This section provides a new method to identify in situ measurements where aircraft emissions are fresh, older or nondetectable. These three situations are assigned to younger contrails, older contrails (contrail cirrus) and unperturbed cirrus, also called natural cirrus. The goal of this new method is to investigate microphysical and optical properties of these three ice cloud classes. Figure 6 shows the full sequence of the airborne in situ measurements of N ($N_{CAS}$, $N_{CIP}$), $R_{eff}$, NO, cirrus (identified using the method in this Sect.), RHi, latitude and altitude as a function of time between 08:30 and 11:30 UTC. N for both instruments, CAS and CIP ($N_{CAS}$ and $N_{CIP}$) and $R_{eff}$ (Fig. 6a, b) indicate the occurrence of cirrus at the flight levels between 10 km and 13 km (Fig. 6g). Many peaks of N up to or above 1 cm$^{-3}$ have been measured. These values are not characteristic for young contrails, where N reaches up to > 1000 cm$^{-3}$ in the first seconds/minutes after formation (Schröder et al., 2000; Kleine et al., 2018) or dilutes to 100 to 300 cm$^{-3}$ after 5 to 10 min (Voigt et al., 2010; Voigt et al., 2017). The contrails probed here are more than 1 h 30 min old, according to the dilution equation of Schumann et al. (2017) under the simple assumption of 1000 particles/cm$^3$ when emissions past the aircraft engine for 1s. Schröder et al. (2000, their Table 2) also show N ~10-20 cm$^{-3}$ for





contrail cirrus older than 30 min. Unterstrasser et al. (2017a) simulate mean N of 20 cm$^{-3}$ after 5 min (labelled as t = 0 h in their paper) falling below 1 cm$^{-3}$ after 2 h of lifetime. Finally, Bock and Burkhardt (2016) find N < 1 cm$^{-3}$ in a contrail cluster after 5.5 h lifetime. Furthermore, the fact that some contrails are also visible in the low-resolution MSG/SEVIRI satellite images (Fig. 2) is another indication that these persistent contrails are at least 1-2 h old (Vázquez-Navarro et al., 2015). Figure

6c shows NO concentration which is produced during the combustion of kerosene. It can also be observed in the UT after fast transport from the boundary layer through convection or in lightning or wildfires. Measurements of NO have been performed continuously inside clouds and also during the lidar legs, where HALO was flying higher in the lower stratosphere (LS). Figure 6c thus shows variable NO concentrations and various peaks of different heights. Since other sources of NO are unlikely in this situation (no thunderstorms in the previous 12 h, no wildfires), we assume that NO concentration increases correspond to

aviation exhausts, with of course peak height differences being caused by various factors such as dilution and aircraft type (Voigt et al., 2010; Jurkat et al, 2011; Jeßberger et al., 2013; Schumann et al., 2013). In fact, after the emission, NO is mixed with the surrounding air and the dilution of this gas increases with plume age such that its measured mixing ratio can be used as a rough indication for contrail age, with high concentrations indicating younger plumes. We exploit these considerations about NO combined with the fact that initial ice particle concentrations N are high due to the high number of soot particles

emitted by the engine (Schlager et al., 1997; Kleine et al., 2018; Bräuer et al., 2021a) when a contrail is formed, but N decreases due to dilution (Schumann et al., 1998) and further processes (Bier et al., 2017; Unterstrasser et al., 2017a) in the course of its temporal evolution. High NO emissions without coincident ice crystal observations correspond to situations where the Schmidt-Appleman criterion is not satisfied and no contrail is formed. This study, similarly to Voigt et al. (2010), uses N and NO as criteria to classify ice crystal observations in the following way.




**Figure 6: In situ measurements of HALO on 26 March 2014 over North Atlantic region, including (a) N for particles larger than 3 µm, (b) R$_{eff}$, (c) NO and NO background, (d) cirrus classification, (e) RHi, (f) flight latitude, (g) altitude.**



First, we determine dynamically the NO background concentration $NO_{background}(t)$ at a given time t by taking the minimum NO value in a 60-second sliding window around this time (Fig. 6c):

$$NO_{background}(t) = min_{-30s \leq \Delta t \leq 30s} NO(t + \Delta t) \qquad (1)$$

$NO_{background}(t)$ takes care of the natural variability of NO in this situation, since other NO sources that could produce NO peaks are excluded here (see beginning of this section). The difference between $NO_{background}(t)$ and the NO(t) curve at each time t is termed $NO_{peak}(t)$

$$NO_{peak}(t) = NO(t) - NO_{background}(t) \qquad (2)$$

and describes the deviation of the NO curve from the background or the NO peak height above the background. NO is thus in this classification as the tracer for aircraft emissions. In a further step we consider ice number concentration N to classify contrails. However, when using N we have to consider that in the lower left portion of Fig. 5a no data is available. Thus, we consider only the right part of Fig. 5a where the full range of $R_{eff}$ could be measured. In doing so we lose all cirrus clouds (usually natural cirrus) with higher $R_{eff}$ (> 30 µm) and small N (< 0.03 cm⁻³) in the upper part of Fig. 5a and also those cirrus

clouds with low $R_{eff}$ and low N (no data in Fig. 5a).

In accordance with Table 1, we first distinguish measurements in cirrus from "1. outside cirrus" when $N_{CAS}$ and $N_{CIP}$ are zero in step 1, and then separate "3. contrail cirrus" and "4. contrails" from "2. natural cirrus" when NO is higher than the background and $N_{CAS}$ or $N_{CIP}$ is larger than 0.03 cm⁻³ (see discussion of Fig. 5a and Fig. 6a) in step 2, reflecting the impact of air traffic. Contrail cirrus (older contrails) is separated from younger contrails by assuming a stronger NO footprint (lower

dilution) in contrails. The rest is labelled as "5. unclassified cirrus" in step 3. This class contains on one side those clouds mentioned above that we cannot classify and likely consists mainly of natural cirrus. On the other side, ice cloud measurements with high $NO_{peak}$ (> 0.14 ppbv) but moderate N ($N_{CAS} \leq 0.4$ cm⁻³) do also fall in this category and should represent observations in younger plumes where few ice crystals could form (maybe because temperature is close to the Schmidt-Appleman criterion) or because ambient air is subsaturated leading to the evaporation of a considerable fraction of newly formed (small) ice crystals.

Summarising, natural cirrus is identified when ice crystals are present with either $N_{CAS}$ or $N_{CIP}$ larger than 0.03 cm⁻³ and NO close to the background value, i.e. $NO_{peak} \leq 0.02$ ppbv; contrail cirrus is characterised by moderate values of NO, i.e. 0.02 ppbv < $NO_{peak} \leq 0.14$ ppbv, and $N_{CAS}$ or $N_{CIP} > 0.03$ cm⁻³; contrails are assumed to consists of many small ice crystals, $N_{CAS} > 0.4$ cm⁻³, and high NO peaks, i.e. $NO_{peak} > 0.14$ ppbv. The remaining cirrus is denoted as "unclassified cirrus".

**Table 1: Cirrus classification according to microphysical properties (N), and tracer measurements (NO) measured by instruments**
**aboard HALO.**

| | Designation | N / cm⁻³ | $NO_{peak}$ / ppbv | Note |
|---|---|---|---|---|
| 1. | Outside cirrus | $N_{CAS} = 0$ and $N_{CIP} = 0$ | Any value | Step 1 |
| 2. | Natural cirrus | $N_{CAS} > 0.03$ or $N_{CIP} > 0.03$ | $NO_{peak} \leq 0.02$ | |
| 3. | Contrail cirrus | $N_{CAS} > 0.03$ or $N_{CIP} > 0.03$ | $0.02 < NO_{peak} \leq 0.14$ | Step 2 |
| 4. | Contrails | $N_{CAS} > 0.4$ | $NO_{peak} > 0.14$ | |
| 5. | Unclassified cirrus | | The rest | Step 3 |

The threshold values for $NO_{peak}$ (0.02 ppbv and 0.14 ppbv) have been determined from the $NO_{peak}$ distribution. $NO_{peak}$ distribution decreases from 0.0 to 0.48 ppbv, with the largest bins being the first and the second one (0<=$NO_{peak}$<0.02 ppbv). Thus, we consider NO to be close to the background value when $NO_{peak}$<=0.02 ppbv and to contain additional NO when $NO_{peak}$>0.02 ppbv. The NO occurrences sink by two order of magnitude (from $10^3$ to $10^1$) until $NO_{peak}$ equals 0.25-0.30 ppbv.

After these values, a tail of relatively seldom high $NO_{peak}$ values (occurrences~$10^1$) is observed. In order to distinguish old





emissions from younger ones, we select the threshold of $NO_{peak}=0.14$ ppbv, in the middle of the range mentioned above. Thus, $NO_{peak}$ between 0.02 and 0.14 ppbv are assumed to capture only older emissions, leaving high NO peaks as well as moderate ones in the range $NO_{peak}>0.14$ ppbv (representing the highest 5% $NO_{peak}$). For the determination of N thresholds for contrails we proceed similarly. In particular, $N_{CAS}<0.03$ cm$^{-3}$ contains an order of magnitude more observations than the next one. Then,

$N_{CAS}$ occurrences decrease by more than 2-3 orders of magnitude from 0.03 to 0.78-0.84 cm$^{-3}$. Again, we take a value in the middle (0.4 cm$^{-3}$) as separation between high $N_{CAS}$ peaks attributable to contrails and moderate values of $N_{CAS}<0.4$ cm$^{-3}$ that we assign to older contrails/contrail cirrus. Please notice that N and NO are selected independently. Notably, these values are just valid for this flight sequence over the NAR on 26 March 2014 during the ML-CIRRUS campaign. In total, from 08:30 to 11:30 UTC we have classified 49 contrail observations, 1018 contrail cirrus observations and 2342 natural cirrus observations

in Fig. 6d from in situ measurements. Unclassified cirrus encompasses 2472 cases, with 94 of them having $NO_{peak}>0.14$ ppbv. This means that at least 18% of all measurements were in contrails/contrail cirrus, confirming the indications gained from MSG and WALES in Sect. 3.1 about the large amount of contrails. We finally remark that MSG/SEVIRI satellite observations are left unused for this classification since the distinction between contrail cirrus and natural cirrus from satellite observations is inherently difficult due to the typical characteristics of young contrails - large N - cannot be measured by passive sensors.

As an example, we first illustrate a contrail encounter between 08:43 and 08:45 UTC (Fig. 7), where we observe contrail properties and ambient atmospheric conditions at a flight level of ~12 km. Related parameters include $N_{CAS}$, NO, aspherical fraction, RHi, $R_{eff}$, and altitude. Close to 08:43:45 UTC, $N_{CAS}$ and NO in the top panel are simultaneously enhanced (and exceed the contrail threshold values presented above) while $R_{eff}$ (around the cloud top, Fig. 3) in the bottom panel decreases to values smaller than those of surrounding ice crystals. According to RHi in the range of 80% and 90% in the middle panel,

we speculate that the rather spherical contrail particles which had formed near water saturation in the exhaust shortly after engine exit, hence at high ice supersaturations have not grown much and are now in the sublimating phase due to the unavailability of gas phase water for deposition. Therefore, they keep their original quasi spherical form and appear to be more spherical than the ice crystals in the surroundings. A prerequisite has to be assumed here that ice crystals develop the complexity at the surface after their initial formation and subsequent growth, and then the complexity disappears during

sublimation, releasing a smooth and near-spherical ice particle, similar with quasi spherical ice particles in convective clouds (Järvinen et al., 2016). Therefore, the reduced aspherical fraction (< 0.5) measured by CAS-DPOL in the subsaturated environment is also an evidence to indicate that contrail particles retain their quasi spherical form if they cannot grow by water uptake and sublimate. In addition, the fact that aspherical fraction in the surroundings reaches up to 1 and that $R_{eff}$ does not exceed 20 μm (which is a relatively small value for natural ice crystals) could be an indication of (natural) ice particles inside

dry layers close to cloud top sublimating next to young contrails that cannot grow larger due to low water vapour concentrations.

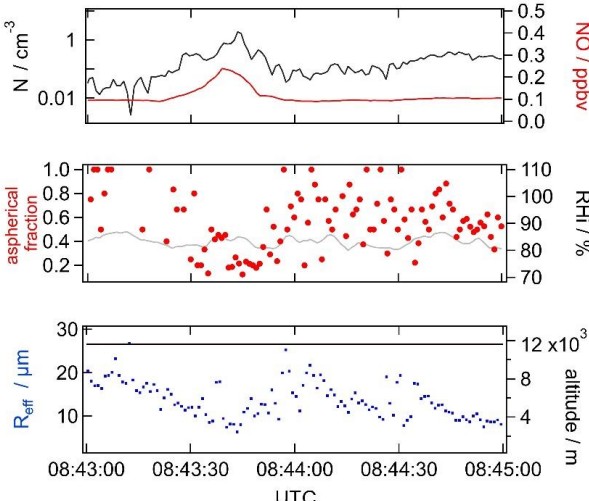

**Figure 7: N, NO, aspherical fraction, RHi, R_eff, and altitude are shown for a short sequence of a dispersed contrail encounter between 08:43 and 08:45 UTC. NO and N are simultaneously referred to identify cirrus classification (top panel). An aspherical fraction value indicates whether the contrail ice particle has a quasi-spherical habit, and RHi is a criterion for saturation. With these two parameters, growth by water vapor uptake and sublimation of the particles are evaluated (middle panel). The size and location of contrails and surrounding cirrus clouds are presented by R_eff and altitude (bottom panel).**

Here, we must note that the decreasing $R_{eff}$ in the subsaturated environment from HALO are solely for this portion of the flight leg (compare to Fig. 2 for an approximate idea about the position of the aircraft to that time). This case study also emphasizes the importance of RHi accuracy, because ice crystal growth and sublimation are determined by ambient humidity conditions in the contrail cirrus regime.

### 3.2.2 Optical thickness and cloud top height

Figure 8a represents the temporal evolution of IOT from CiPS. In the northern part (Fig. 6f) of the HALO race track, IOT is between 0.05 and ~0.3, sometimes reaching up to almost 0.4. In the southern part (i.e. at ~8:50, 9:30, 10:25 and 11:05 UTC) IOT is larger with peaks up to 0.6 and 0.8, corresponding to the thicker cirrus already noticed in Fig. 2. CiPS indicates a mean IOT of around 0.2 over the entire flight with a variability of ±0.1. Further, we order MSG/SEVIRI pixels according to the classification in Sect. 3.2, which is performed using the in situ data collected at given height levels in the cloud, and produce statistics for them separately. This attribution of course is not unambiguous since at different heights cloud properties the HALO flight track can be different. Following the discussion of the lidar data in Sect. 3.1, for instance the fall streaks of the contrails – containing larger crystals than the contrail cores - will also contribute to the contrail IOT and the upper layer cloud, whose origin is unclear – will add up to almost all measurements. Thus, it does not surprise that the median IOT for the three categories "natural cirrus", "contrail cirrus" and "contrails" are almost identical (Fig. 9a). The average IOT of natural cirrus and contrail cirrus are 0.21 and 0.24 respectively, and 0.15 for contrails. Furthermore, contrail pixels (which are much less in number than the other two categories) have a much smaller variability, with IOT always < ~0.3, while IOT for contrail cirrus and natural cirrus can be much higher. Fresh contrails are expected to have a higher IOT than contrail cirrus (see e.g. Unterstrasser and Gierens, 2010; Iwabuchi et al., 2012; Schumann et al., 2017) because of their large number of small ice crystals. Also, in the case of contrail formation within natural cirrus the increase in ice crystal numbers should lead to an increase in optical thickness right after formation. Comparing to literature, optical thickness of young contrails is estimated to be around 0.24 by Minnis et al. (2004), 0.240-0.274 by Iwabuchi et al. (2012), 0.216 (Bedka et al., 2013), 0.147±0.120 (Minnis et al., 2013), all from polar orbiting instruments with better spatial resolutions. From MSG/SEVIRI, Vázquez-Navarro et al.





(2015) show that the probability distribution function of "young" contrails (young with respect to the MSG/SEVIRI spatial

resolution: 30 min old at most after their first appearance in MSG/SEVIRI images) peaks at IOT < 0.2 and that IOT increases

by ~ 0.1 after 3-5 h, leaving only either sublimating contrails pixels after 8 h or thicker cirrus pixels (IOT~0.4).

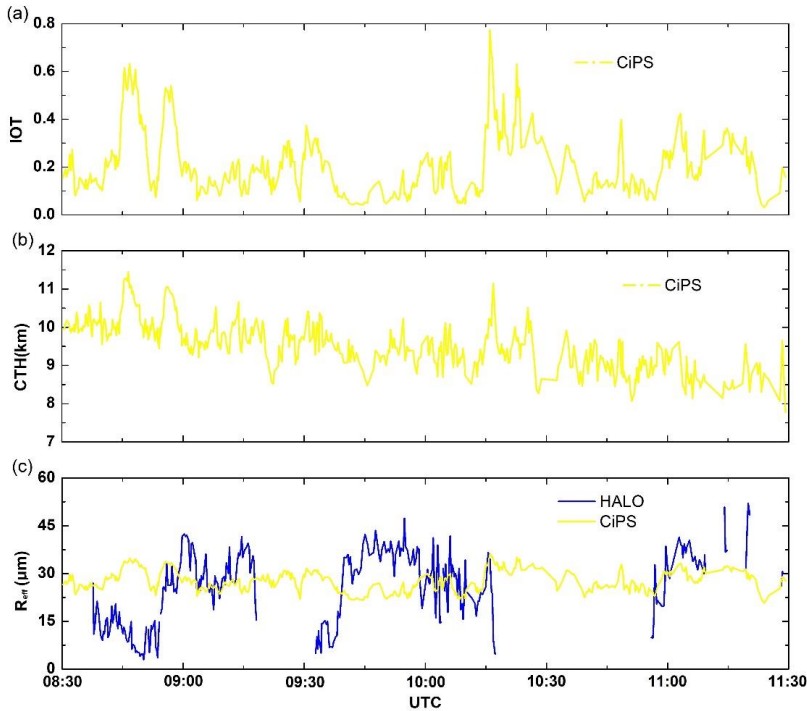

**Figure 8: Temporal evolution of (a) IOT from CiPS, (b) CTH from CiPS and (c) $R_{eff}$ from CiPS and from HALO at SEVIRI spatial**
**resolution.**

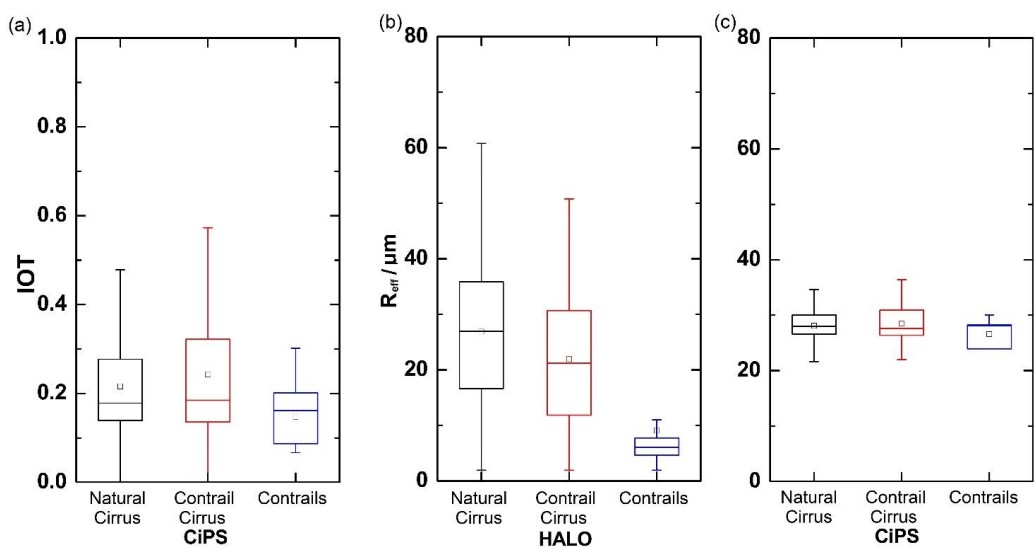

**Figure 9: Box charts of (a) IOT from satellite observations by CiPS, (b) $R_{eff}$ from HALO in situ measurements, and (c) from CiPS. Each bar contains the information of the maximum, minimum, interquartile range (25% percentile to 75% percentile), median (horizontal line) and mean (square).**



Minnis et al. (2013) also show an increase of IOT during the temporal evolution of contrails to contrail cirrus (IOT=0.431, 3 times as large as the IOT of young contrails). In contrast, Unterstrasser and Gierens (2010) simulate a rapid decrease in optical thickness in the first 20 min after formation from values up to 0.7 to ~0.2, followed by a slow decrease of IOT with time. In Unterstrasser et al. (2017a) optical thickness also decreases fast during the first hour after contrail formation to IOT around 0.05-0.2, independently of synoptic properties, but afterwards cases of moderate updraft induce an increase in IOT again. In

their compilation of contrail properties, Schumann et al. (2017) also indicate a decrease from IOT~1.0 to IOT~0.1 for contrails of age between 10 and 100 s, with IOT remaining almost constant up to 10,000 s (2 h 45 min). In simulations, Bock and Burkhardt (2016, their Fig. 7) show that ice crystal numbers can decrease while IWC increases and optical thickness – which is a vertically integrated quantity – does not have to necessarily decrease. Moreover, Bier et al. (2017, their Fig. 4g) show that in certain cases the mean optical thickness of a contrail cluster is first increasing. They also stress that there is a large variability

in the temporal evolution that to a large extent depends on the synoptic situation (and not on microphysics). Thus, comparing to this literature, the contrails observed here (Sect. 3.2) are already relatively old and their IOT is comparable with that of contrail cirrus and natural cirrus, but due to the vertically integrated nature of passive satellite observations (and of IOT) differences between the classes are to some extent washed out. From the point of view of optical thickness, the entire cloud seems to be homogeneous without remarkable differences among the cloud types defined in Sect. 3.

With respect to CTH in Fig. 8b, CiPS results show that the cloud gradually decreased from the average height of 10.5 km at 08:30 UTC to ca. 8.5 km at 11:30 UTC, which present qualitatively the same pattern as lidar legs in Fig. 3 observed, although the CiPS values are lower by ca. 1.5 km in this case. The optically thicker cirrus cloud parts observed in the South (see above) are associated to a slightly higher altitude between 11.5 and 11.0 km, thus closer to the altitude observed by the lidar. Both remote sensing instruments, the lidar on HALO and MSG/SEVIRI, observe a sinking motion of the cloud that leads to its

(partial) dissipation after the time period considered here (08:30-11:30 UTC), as can be seen in the IOT plots in Fig. 2.

### 3.2.3 Effective radii

In Fig. 8c a temporal sequence of the in situ and remote sensing $R_{eff}$ from 08:30 to 11:30 UTC on 26 March 2014 is shown. As illustrated in Sect. 3.1, cirrus properties and characteristics can be very different at different heights and positions. In particular, contrails seem to occur at certain levels, probably corresponding to a particular flight level, and at particular times.

Furthermore, HALO was flying at different levels during the in situ legs such that different parts of the cloud could be probed: sometimes maybe the contrail core, sometimes ice crystals sedimenting out of the core, sometimes in even older contrails or in fresh ones. For instance, the first part of the first in situ leg at approx. 11.6 km seems to be most of the time between two cirrus cloud layers or at least in a thinner cloud layer where RHi is below saturation (also see in Fig. 6e, g) and ice crystals are sublimating. This part of the flight is characterised by small $R_{eff}$ (between 5 and 25 µm) and many observations with high N

(above 0.4 cm$^{-3}$) as well as NO (above 0.15 ppbv). In the second part of the first in situ leg, HALO descended to 11 km where many contrails were expected, at least around 52°N (see Fig. 6f) together with ice crystals sedimenting down from the upper layer or with ice crystals from natural or older contrail cirrus. This section is characterised by a mixture of small (3-20 µm) and high $R_{eff}$ (up to 50 µm). The smaller $R_{eff}$ are mainly at ca. 53.2°N, a little bit further North than the bulk of contrails observed in lidar leg 1 (up to 53.1°N) but in a location where some contrails maybe from the upper layer could be measured.

RHi in this part of the first in situ leg is around 100%. The second in situ leg was flying first very close to cloud top, then down to 10.8 km and subsequently ascending in plateaus again to 11.2 km before ascending to 12.9 km. Very close to cloud top N is rather small (max. 0.3 cm$^{-3}$) and $R_{eff}$ as well (mainly below 15 µm), indicating that only small particles are left here and/or that evaporation takes place in subsaturated air (RHi close to 90%). Hence, although in general natural ice nucleation especially close to cloud top is expected, this does not seem to be the case here at the time of the HALO flight where the cloud was

already sinking (but natural ice crystals might have nucleated before or in the portion of the clouds with high RHi). At the second altitude level (10.8 km), in this leg $R_{eff}$ is larger (around 40 µm) but N remains small and RHi is closer but usually





below 100%. Afterwards, during the second altitude step in the ascent of HALO, RHi is well above 100% and $R_{eff}$ is between 25 and 50 µm, with some N peaks above 0.3 cm⁻³. This part of the flight path is between 52 and 54°N, i.e. in the Northern part of the race track where less contrails are observed by the lidar. In the last and highest plateau of the second in situ leg humidity is again close to 100%, $R_{eff}$ become smaller (around 20 µm with large scattering) and many N peaks are observed. When HALO finally climbed to 12.9 km for the last lidar leg, $R_{eff}$ and RHi sink very rapidly in the stratosphere. In the last in situ leg, only the first part is in-cloud at 11 km. $R_{eff}$ and RHi increase from cloud top (subsaturated) to this height, with supersaturation in the second part of the plateau. Towards the end of the plateau (at 51.5°N ca.) some very high N peaks are measured and $R_{eff}$ is around 30 µm. Finally, HALO dived below the ice cloud. In general, as Fig. 6e and Fig. 5b show, the RHi for these 3 in situ legs are both below and above 100%, i.e. in subsaturation and supersaturation. In particular, many ice crystals are found in air with RHi<100%.

Considering the entire flight and separating $R_{eff}$ measurements according to our classification enables us to study the statistical properties of the ice crystal sizes. From Fig. 9b, contrail cirrus $R_{eff}$ probed from airborne instruments are smaller than those of natural cirrus, with an average $R_{eff}$ of about 22 µm. The natural cirrus has an average $R_{eff}$ of 27 µm, which is at the lower end of the particle size distributions observed in natural cirrus (Schröder et al., 2000). The mean radii of contrail cirrus or contrails from in situ measurements is 18% smaller than that of natural cirrus. From contrails to contrail cirrus, the mean and medium values of particle size mainly show an increasing tendency by an average difference of 13 µm (144%). Despite the uncertainty of probed particle sizes, the radii of natural cirrus reached up to 60 µm, while the majority of maximal $R_{eff}$ of contrail cirrus topped at 53 µm, in agreement with findings of Voigt et al. (2017). This is consistent with the physical picture that contrails form initially as small particles and increase in size by water uptake in ambient air, when air is supersaturated. If no supersaturation is present, contrails dissipate and cannot evolve into contrail cirrus. However, contrail cirrus as well as natural cirrus can consist of sublimating particles when air becomes subsaturated (Kübbeler et al., 2011).

Finally, we jointly assess radii variations of natural cirrus and adjacent contrail cirrus from CiPS with simultaneous HALO measurements for this NAR case. Here, HALO in situ $R_{eff}$ is shown as an average over each SEVIRI pixel. However, satellite observations have contributions from the entire cloud column, and in situ measurements are solely for one certain flight level, and in this case, contrails are embedded in ice cloud layers (Fig. 3) that probably have different microphysical and optical properties, consisting either of natural or contrail cirrus. In fact, we have seen above that a difference in $R_{eff}$ is present among the three cloud classes. These reasons hinder a one-to-one comparison between in situ and spaceborne measurements. Nevertheless, we want to investigate whether the thermal satellite observations of CiPS are able to collect some characteristics of the contrails and contrail cirrus. The temporal variability of CiPS $R_{eff}$ along the flight path is smaller than that of simultaneous in situ $R_{eff}$ averaged over the MSG/SEVIRI pixels (Fig. 8c). $R_{eff}$ from the thermal CiPS retrieval varies around ~30 µm along the entire flight path and no clear correspondence is observed. Nevertheless, mean values (Fig.9c) of CiPS are not far away from the in situ values and the $R_{eff}$ for contrails shows a smaller range than those for the other two cloud classes. Of course, this might also be due to the smaller number of contrail samples.

## 4 Radiative effects of contrails, contrail cirrus and natural cirrus

Here, we develop a new estimation method for TOA net instantaneous cirrus RF that relies on satellite and in situ observations. TOA irradiance with contrails/contrail cirrus/natural cirrus can be observed, but the comparable situation without ice clouds has to be obtained from another source, e.g. from the surroundings (like e.g. in Vázquez-Navarro et al., 2015 for contrails) or from model simulations (Haywood et al., 2009 for contrail cirrus). The instantaneous net RF at TOA is the change of the total irradiation under a situation with cirrus minus the irradiances in the same situation without cirrus. In fact, the SW component of the cirrus RF $RF_{SW_{TOA}}$ of the cirrus can be diagnosed as

$$RF_{SW_{TOA}} = SW_{\uparrow no\ cirrus} - SW_{\uparrow cirrus} .$$ (3)





Notice that $SW_{\uparrow cirrus}$ corresponds to the RSR that can be observed by the satellite. Similarly, the LW RF $RF_{LW_{TOA}}$ can be diagnosed from

$$RF_{LW_{TOA}} = LW_{\uparrow no\ cirrus} - LW_{\uparrow cirrus}\ . \qquad (4)$$

Again, the cirrus term $LW_{\uparrow cirrus}$ in Eq. (4) corresponds to the OLR that can be computed from MSG using the RRUMS algorithm. The net cirrus RF $RF_{NET_{TOA}}$ is defined as

$$RF_{NET_{TOA}} = RF_{SW_{TOA}} + RF_{LW_{TOA}}\ . \qquad (5)$$

However, the two values (TOA irradiance with cirrus and without cirrus) are similar in size since contrails and contrail cirrus represent a small perturbation. In fact, both in Vázquez-Navarro et. (2015) and Haywood et al. (2009) taking the difference between the cirrus contaminated and the cirrus-free irradiances leads also to negative values of RF in the thermal range (OLR with ice clouds is larger than OLR without ice clouds) or to positive RF in the solar range (RSR with ice clouds is smaller than RSR without ice clouds). This is of course unphysical and we would like to avoid it. To this end, in Sect. 4.2 we develop a new method based on RTM calculations exploiting airborne measurements, satellite observations and ERA5 model atmospheric data that produces TOA irradiance that is fully consistent for both cirrus and cirrus-free regions. The combination of these datasets however does not allow to take care of the full vertical and spatial variability of clouds, as they can be observed for instance in the WALES data (Fig. 3), and assumptions have to be introduced. So finally, to ensure that TOA irradiance calculated this way is realistic, we compare our RTM simulations of RSR and OLR for the cirrus contaminated case with satellite observations of RSR and OLR from RRUMS at pixel level. An additional comparison on a 3x3 pixel scale with the more accurate GERB products is performed to assess the RRUMS accuracy on this particular day (Sect. 4.1). We don't apply GERB directly since we want to evaluate RF along the HALO path and GERB is available for 3x3 SEVIRI pixel areas.

### 4.1 Evaluation of RRUMS with GERB

To evaluate TOA irradiance from RRUMS during this day, we compare GERB products (RSR$_G$ and OLR$_G$), comprising 3x3 pixels of MSG/SEVIRI, with RRUMS results along the HALO flight trajectory. Notably, different cirrus classifications have the same TOA value from GERB products and RRUMS results since the temporal/spatial resolution of the in situ data is much higher than that of SEVIRI. We first remove 7.5% bright errors and 1.3% cold bias of the GERB irradiances in the solar and thermal range (Clerbaux et al., 2009) to build new revised-GERB datasets. We then calculate the 3x3 pixel mean values of RRUMs results. The corresponding RSR$_R$ correlates well with RSR from GERB (RSR$_G$: correlation coefficient CC=0.72), but RRUMS tends to overestimate GERB (Fig. 10a).

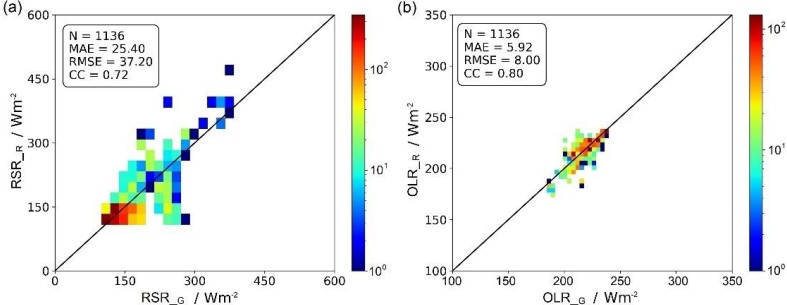

**Figure 10: Comparison of TOA (a) RSR and (b) OLR from RRUMS algorithm results (RSR$_R$, OLR$_R$) and revised-GERB datasets (RSR$_G$, OLR$_G$) (3×3 SEVIRI pixels) along the HALO flight on 26 March 2014. The MAE, RMSE and CC are used as metrics.**

In terms of OLR, the agreement is good, with a mean absolute error around 6 W m$^{-2}$ and a correlation coefficient of 0.80 for OLR$_R$ and OLR$_G$ (Fig. 10b).


**4.2 RTM simulations of TOA irradiance along the HALO flight path**

In this section we illustrate the new method used to derive instantaneous cirrus RF (see introduction of this section). In particular, we explain a 2-step method to compute both cirrus-free and cirrus influenced TOA irradiances. First, an atmosphere is set up as input to the RTM that contains a realistic representation of the situation observed including in particular all clouds. Then in a second step the ice clouds are removed, for which the RF is to be computed. This provides the cirrus-free reference TOA irradiance for the calculation of cirrus RF. This means that the RTM calculations are performed twice, once for the conditions unaffected by contrails and contrail cirrus at the flight level and once with contrail and contrail cirrus. The basic setup of the RTM is described in Sect. 2.3. Every RTM calculation needs an atmospheric state as input. This is in part obtained from ECMWF ERA5 reanalysis data (Hersbach et al., 2020). All 137 model levels are used. The horizontal and temporal resolution are $0.25° \times 0.25°$ and 1 h. We derived temperature profiles, logarithm of surface pressure, specific humidity, ozone mass mixing ratio, and land or sea mask. Densities of gaseous water ($H_2O$) and ozone ($O_3$) are derived from specific humidity and $O_3$. $CO_2$ is set to have a volume mixing ratio of 400 ppm. Vertical profiles of liquid clouds are extracted from ERA5 data as well since it is difficult to determine their presence and properties from SEVIRI when they have a cirrus cloud on top. We choose vertical profiles closest to the HALO flight time, i.e. with a time difference of 30 min at most between model and in situ measurements. We rely on the fact that reanalysis data should provide a realistic description of cloud properties and cloud positions, but we accept that this procedure might cause small temporal shifts such that observations close to cloud edges are not represented in an optimal way through the model data. However, we refrain from interpolating cloud properties in time since also this procedure would create artificial clouds that do not exist in reality, especially in locations where no cloud is present at a given time but it is there at the next time. For liquid clouds, the parameterization by Bugliaro et al. (2011, 2022) are applied for creating $R_{eff}$ profiles and optical properties from model output. Of course, their presence and properties are crucial especially to RSR and the situation encountered here is particularly challenging since the HALO track (Fig. 2) runs almost parallel and sometimes very close to the edge of a low level (yellowish in the RGBs) cloud field. The solar zenith angle corresponds to synchronous SEVIRI observations. Besides, we used default values of 0.025 or 0.975 for ocean surface broadband albedo or emissivity respectively, as they are in the average range. For ice clouds, another procedure is applied. Since SEVIRI observations with CiPS are able to account for the entire cirrus cloud layers but are not affected by low lying clouds, SEVIRI provides accurate ice cloud properties (IOT) that can be used in the RTM.

The representation of cirrus at the flight level is complemented by adding their $R_{eff}$ from in situ measurements, CTH and cirrus bottom height (CBH) from lidar legs, as well as CiPS IOT into libRadtran in the way described in the next lines. This way we simulated a vertically homogeneous ice cloud with the correct IOT obtained by CiPS. We assumed IOT to be constant over an entire SEVIRI pixel while $R_{eff}$ varies according to in situ (however at the single altitude levels probed by HALO). Since the RTM needs IWC as input, we determine it from IOT. First, Extinction *Ext* for IWC = 1 g m$^{-3}$ for each measurement is first interpolated to the given in situ $R_{eff}$ with the parameterisation of optical properties of Baum et al. (2011) for rough aggregates. The selection of this shape for all cloud types is motivated by the fact that each cloud column, even those containing with contrails, encompass ice crystals with various temporal evolution, e.g. young small – probably round - ice particles in contrail cores together with larger sedimenting ice crystals in the fall streaks with different shapes or with evaporating ice crystals – in part still large in size - in undersaturated air that are starting to lose asphericity or maybe even natural ice crystals produced by homogenous nucleation with unknown shape. To avoid an additional arbitrary choice with respect to ice particle shape we decided to keep the method as simple as possible and selected this shape (rough aggregates) as e.g. for the MODIS optical property products Collection 6. Note the parametrization of this shape only covers $R_{eff}$ from 5 to 60 µm, which results in the inexecutable RTM calculations for larger or smaller ice crystals. The IWC for each measurement of $R_{eff}$ corresponding to a vertically homogeneous ice cloud with given IOT is derived using the following equation:

$$IWC = \frac{IOT}{Ext*(CTH-CBH)} \tag{6}$$





CTH and CBH are obtained from the lidar legs since they seem to vary only slowly with time (Fig. 3). Thus, this IWC is used to simulate a homogeneous ice cloud layer between CBH and CTH and corresponds to the IOT observed by CiPS.

Finally, every in situ measurement of $R_{eff}$ is assigned to a cloud class (Sect. 3.2.1) and a collocated IOT from CiPS as well as
the ERA5 properties listed above for temperature, gas and liquid water clouds. With this atmospheric setup, TOA irradiances are computed and represent the cirrus contaminated RSR and OLR. Then, the ice clouds only are removed from the input of the RTM and other calculations are performed to compute cirrus-free irradiances. Both together are then inserted in Eqs. (3)-(5) to compute the instantaneous net RF of the cirrus cloud under consideration.

The RSR and OLR values influenced by contrails, contrail cirrus and natural cirrus as obtained from libRadtran simulations
along the flight path of HALO are presented in Figure 11 (RSR$_L$ and OLR$_L$) together with the corresponding values computed with the RRUMS algorithm (RSR$_R$ and OLR$_R$) as reference. The two methods agree quite well, but the RTM calculations tend to underestimate large RSR values (at 9:15, 10:15 and 11:10 UTC), probably due to too thin, missing or even mismatched liquid water clouds that are taken from ERA5. Uncertainties in ice cloud properties cannot have such a large effect on RSR since their optical thickness is very low (Sect. 3.2.2). Furthermore, a slight overestimation of RSR by the RTM compared to
RRUMS during leg 2 (9:30-10:30 UTC) is also observed, maybe related to an underestimation of ocean albedo during this time. For OLR, the agreement between the two datasets is good, with both slight (< 10 W m$^{-2}$) overestimations and underestimations.

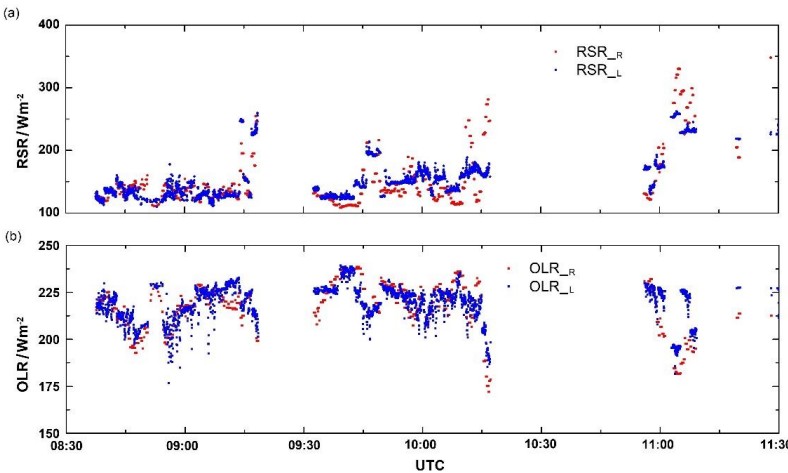

**Figure 11: TOA RSR and OLR from our RTM simulations (RSR$_L$, OLR$_L$) for probed ice particles and RRUMS algorithm results**
**(RSR$_R$, OLR$_R$) for single SEVIRI pixel along the HALO flight path. All simulations are plotted here.**

The RTM based TOA radiation estimation method provides a good representation of the real atmosphere and will be used to determine RF of the ice clouds probed during this flight in the following section. However, to reduce the impact of the incorrect consideration of low level clouds mentioned above, an additional filtering is used. We consider the ratio of the RMSE value of RSR from RRUMS against GERB (Sect. 4.1) divided by the mean RRUMS RSR (ratio=0.19) as a measure for the
645 uncertainty of RRUMS and neglect all RTM simulations that differ by more than this fraction from RRUMS. This way, 1724, 795 and 19 points for natural cirrus, contrail cirrus and contrails are left, with 191, 79 and 0 points filtered out for each category.

**4.3 TOA radiative forcing during the HALO flight**

We evaluate RF of probed contrail cirrus and natural cirrus along the HALO flight route using Eqs. (3)-(5). TOA cirrus-free RSR and OLR along the flight path of HALO are calculated by excluding the ice cloud layer at the flight level from the
650 radiative transfer calculations. Figure 12a indicates the patterns of TOA RF for natural cirrus, contrail cirrus and contrails (around the flight level) versus time. Strikingly, natural cirrus and contrail cirrus show very similar patterns in SW, LW and



net RF due to their similarity in IOT and despite small differences in $R_{eff}$. Both positive and negative net RF values are present with largest variability before 9:45 UTC. This is due to the peaks of higher IOT up to approx. 0.6 (Fig. 8) that induce a higher reflection of solar radiation and also make the cloud less transparent to thermal radiation thus increasing both its SW and LW forcing. However, in case of an ice cloud on top of a low-level cloud, especially SW RF is usually reduced, since the increase of reflection is then less pronounced. From 9:30 to 10 UTC net RF is mainly positive and after 10 UTC a tendency to negative net RF is observed. The larger values of RF around 10:15 UTC are again caused by the increase in IOT. After 10:15 UTC HALO was then performing the third lidar leg (Fig. 6) such that they are not included in this calculation. Also, the few contrails show either slightly positive ($< 5$ W m$^{-2}$) values or negative values ($> -2$ W m$^{-2}$). Over the entire flight, CTH is sinking such that earlier observations should correspond to slightly higher OLR (cloud top is sinking only slowly, Fig. 3) and thus larger LW RF since sea surface temperature, that largely affects cloud-free thermal irradiances, is staying almost constant over the day. However, this effect is modulated by cirrus IOT and by the presence of low-level clouds (as explained above).

As depicted in Fig. 12b, the average natural cirrus RF in Fig. 12a are 7.7 and -7.1 W m$^{-2}$ in the LW and SW, respectively, resulting in a net forcing of 0.6 W m$^{-2}$. Contrail cirrus net RF of 1.6 W m$^{-2}$ is usually larger than that of natural cirrus, with the LW and SW RF of 8.8 and -7.2 W m$^{-2}$. As for contrails, their RF in the LW and SW are 6.2 and -5.8 W m$^{-2}$, with a net value of 0.4 W m$^{-2}$. Thus, RF from contrail cirrus is a factor of 4 higher than RF for contrails, which is mainly due in these RTM simulations to the lower IOT of contrails (Fig. 9a).

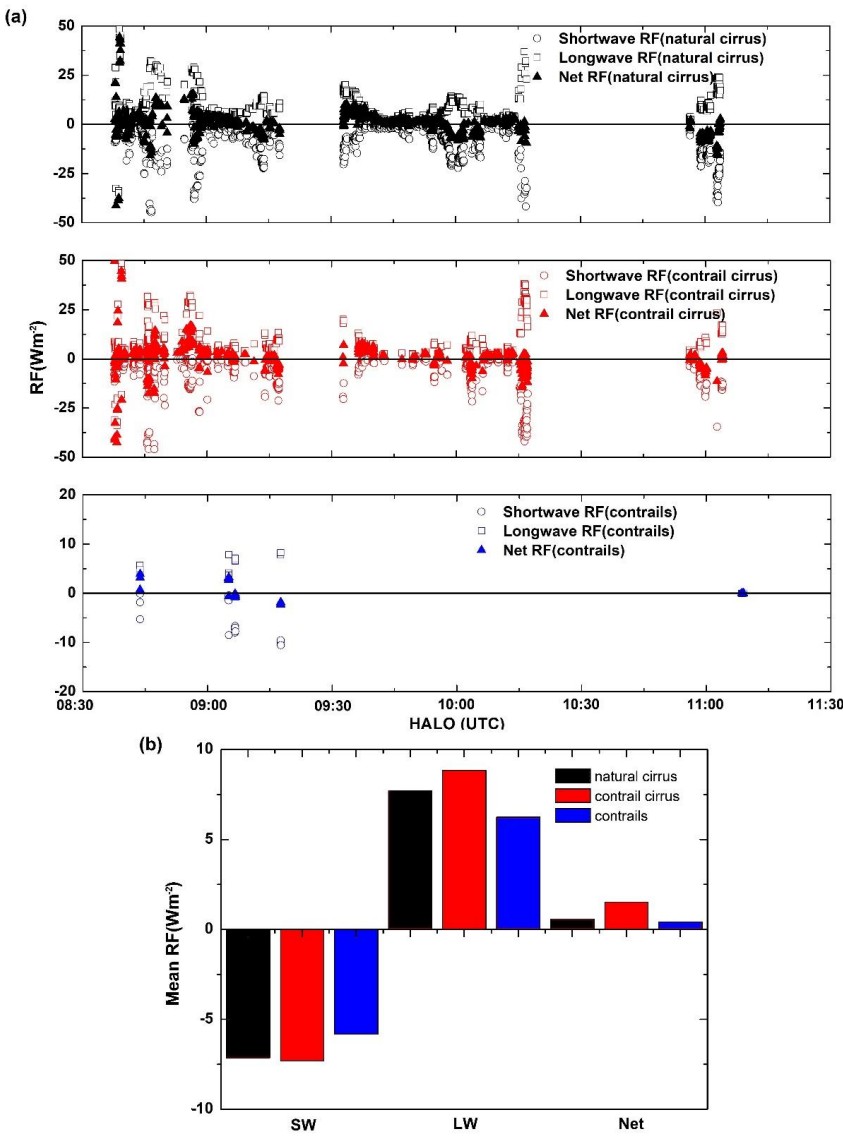

**Figure 12: (a) TOA LW, SW and net RF of natural cirrus, contrail cirrus and contrails from our RTM simulations for HALO probed ice particles along the HALO flight path over NAR on 26 March 2014, and(b) corresponding mean values.**

### 4.4 Diurnal cycle of TOA RF of ice clouds in the area of the HALO flight

In Sect. 4.3 we presented TOA RF for each waypoint of the probed cirrus. In order to examine whether microphysical properties and radiative effects of cirrus detected during the HALO flight are representative of this area and this time and analyse the corresponding temporal variation, we present the cirrus cover (CC), cirrus IOT and CTH (all three quantities from CiPS), ambient atmospheric conditions (vertical velocity and RHi from ERA5) and RF in an area enclosed by a red box in Fig. 2 near the HALO flight track. This area extends mostly West of the HALO flight thus containing the eastern edge of the ridge cloud or at least the contrails/contrail cirrus formed directly East of the ridge cloud. In the East the red area is already at 8 UTC partly free of ice clouds. The classification of ice clouds into different classes as in Sect. 3.2.1 is of course not available such that we consider here all ice clouds in the region. Considering the satellite observations in Fig. 2, we suppose that the ice cloud field is impacted by aviation in a similar way as the HALO flight path.





RF in this section is at the SEVIRI resolution, and calculated using CTH, IOT and $R_{eff}$ from CiPS, and CBH estimated from CiPS and cloud thickness from WALES assuming the latter doesn't change in the area with the same procedure presented in Sect. 4.1 based on the RTM libRadtran. Figure 13a, b, and d show the regional CC, average IOT and CTH from CiPS, and

mean TOA SW, LW and net RF for an area of $36 \times 51$ pixels (~16500 km$^2$) computed with libRadtran. Fig. 13c represents the mean vertical velocity and mean RHi within the pressure difference of $\pm 25$hPa around 225 hPa (the average pressure height of HALO) in the area of interest.

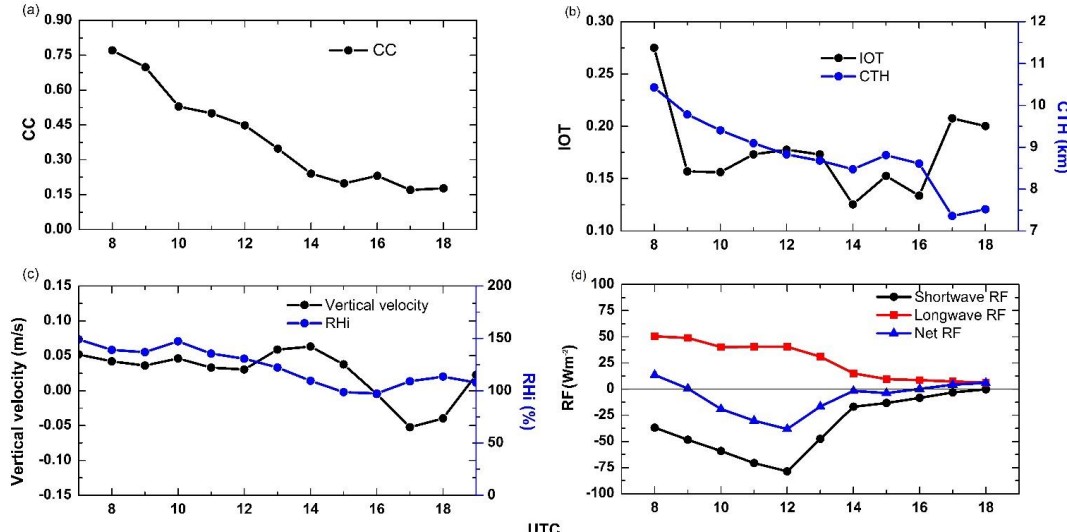

**Figure 13: The variations of (a) CC, (b) mean IOT and mean CTH, (c) mean vertical velocity and RHi within the pressure difference**
**of 25hPa around 225hPa, and (d) SW, LW and net RF within the area indicated by a red box in Fig. 2.**

In Fig. 13a, we observe that CC gradually decreases from 0.75 at 08 UTC to 0.15 at 18 UTC (this can be observed in Fig. 2 until 12 UTC) with the steepest decrease until 14 UTC (CC=0.25) and a slower decrease from 14 to 18 UTC. The positive vertical velocity before 15:00 UTC shown in Fig. 13c implies the local downward motion of airmass to warmer temperature layers and leads to the reduction of RHi. Similar to Sect. 3.2, IOT in Fig. 13b decreases between 08:00 (0.28) and 10:00 UTC

(0.15), then slowly increases until 12 UTC (0.18), then falls below 0.15 until 16 UTC and finally goes back to 0.20. CTH decreases during the day and is thus consistent with both the observations of HALO (Fig. 3) and the downward motion (Fig. 13c). Mean CTH values over the red area are similar to those along the HALO flight track (Fig. 8). Since an underestimation of CTH by CiPS with respect to WALES (Fig. 3) is observed there, we might assume that CTH is also underestimated by CiPS in the red area.

Similar to and at some time unlike the situation along the HALO path, mean net RF over this area in this synoptic situation is positive in the early morning until 9 UTC and in the afternoon after 16 UTC, although very close to 0 W m$^{-2}$. Of course, after 14 UTC CC is already very low such that the mean RF values also tend to approach 0 W m$^{-2}$. The strongest cooling is at 12 UTC and the maximum of net RF (warming) is at 08:00 UTC. From 9 to 16 UTC net RF is mainly negative and thus this ice clouds with the contrails and contrail cirrus tend to cool during daytime. Notice however that the possible underestimation of

CTH by CiPS in this area would result in the general underestimation of the LW RF results since a lower CTH reduces the contrast to the cirrus-free OLR. In turn, this would further shift cirrus net RF towards cooling.





## 5 Summary and conclusions

This study provides a detailed investigation of a situation with contrails, contrail cirrus and natural cirrus that combines airborne in situ probes, airborne lidar measurements and geostationary satellite observations. We have tackled aspects related to microphysical properties and radiative effects through one case study focusing on the NAR on 26 March 2014 during the ML-CIRRUS experiment. On the morning of that day the HALO research aircraft flew for three hours in or above cirrus clouds with contrails that were formed probably some hours before. Various contrails with related fall streaks can be identified in

airborne lidar backscatter images and merge to a larger cloud with a vertical extent of approx. 2 km that dissipates with time. BTDs from MSG/SEVIRI also reveal the presence of various line shaped structures that represent persistent contrails. Simultaneous airborne humidity measurements show ice subsaturated regions in large parts of the cloud, but also large supersaturations (RHi > 140%) that would also support natural ice cloud formation events. We assume that contrails formed, evolved and merged and that homogeneous nucleation of natural ice particles followed.

Primarily, a "contrails, contrail cirrus and natural cirrus" database has been set up with a novel method from airborne measurements that identifies aircraft plumes from NO trace gas measurements and differentiates contrails from contrail cirrus on the basis of ice number concentrations at the flight level. Along the HALO flight path, MSG measured an average IOT for the cloud layers containing the natural cirrus, contrail cirrus and contrails (classified from in situ) of 0.21, 0.24 and 0.15, respectively. Fresh contrails are expected to have a higher IOT than contrail cirrus due to their large number of small ice

crystals. However, contrails probed here are already at least 1 h old and during their evolution the deposition of ice particles competes with the fast dilution of the plume. Thus, observed optical thickness of three cloud types are quite homogeneous, while still a difference in $R_{eff}$ inside the cloud is present. The mean radii of contrail cirrus or contrails from in situ measurements is about 18% smaller than that of natural cirrus. As for microphysical variations from contrails to contrail cirrus, particle sizes increase by about 114% by deposition of ambient water vapor. In fact, RHi correlates with effective ice crystal size, with

smaller $R_{eff}$ for RHi close to or below saturation and larger $R_{eff}$ in ice supersaturated air. As far as MSG observations of $R_{eff}$ in the thermal range with the CiPS algorithm are concerned, mean values are close to the in situ values and the $R_{eff}$ derived from the size distribution of contrails shows a smaller range than those for the other two cloud classes. However, it seems that due to the sensitivity of the thermal algorithm and the complicated vertical structure of the cloud, CiPS is not able to reliably identify the presence of contrails (with smaller $R_{eff}$) in the cloud layer.

For purpose of obtaining accurate radiative effects of natural cirrus, contrail cirrus and contrails, a new TOA RSR and OLR estimation method is developed, which is based on detailed RTM calculations and exploits in situ measurements, satellite observations and ERA5 model atmospheric data. Using IOT from MSG/SEVIRI (CiPS), $R_{eff}$ from in situ and cloud top and bottom height from the lidar, and gas, temperature and liquid water cloud profiles from ERA5, an input atmospheric state for the RTM has been defined that enables to compute RSR and OLR that compare well to MSG measurements (RRUMS). When

the ice cloud layer (contrails, contrail cirrus, and natural cirrus) is removed from the RTM input, RSR and OLR for cirrus-free conditions can be computed that are consistent with the corresponding cirrus contaminated quantities. This way, RF for the three cloud classes identified in situ can be computed. Contrail cirrus net RF is larger by a factor of 4 with respect to contrails RF and is larger than that of natural cirrus as a whole. However, we have to note that few individual contrails observed here have a smaller RF.

For a larger area of 36 × 51 SEVIRI pixels adjacent to the HALO flight path, the diurnal cycle of cirrus (all types together) was computed. RF of cirrus is associated with the changes of CC, CTH and IOT. Here, the positive net RF lasts till 9 UTC and then during daytime the mean net RF of cirrus, contrail and contrail cirrus becomes negative, leads to a reduced energy input into the atmosphere and to a cooling effect. In hourly resolved simulations of the contrail cirrus RF in the Northern Atlantic Flight corridor in 2019, Teoh at al., (2022) also find many cases of contrails and contrail cirrus cooling during daytime, while

contrail cirrus of course warms the atmosphere during nighttime. We present only observations of one day in one particular synoptic situation. It is known that synoptic conditions have a large impact of contrail cirrus properties (Bier et al., 2017).



Nonetheless, according to Muhlbauer et al. (2014) almost half of the cirrus cloud occurrences close to the Atmospheric Radiation Measurement Southern Great Plains (SGP) site in Oklahoma during the Department of Energy Small Particles in Cirrus (SPARTICUS) campaign was explained by three distinct synoptic regimes, one of them being upper level ridges.

Assuming that this is true for cirrus clouds in European mid-latitudes as well (despite the fact that SGP is located of course over land and close to 36°N, which corresponds to the southern tip of the Iberic Peninsula in Europe and thus has a warmer climate than the NAR) and together with the fact that air traffic to and from the US takes place every day, the situation encountered here might take place often in this area and our numbers could be representative for such conditions. In contrast, since frontal clouds are expected to be optically thicker, contrails and contrail cirrus formed in connection with such systems

are probably not comparable to the one presented here.

This work is valuable for identifying natural cirrus, contrails and contrail cirrus, estimating TOA RF, assessing microphysical properties and climate impacts of natural cirrus and anthropogenic cirrus, and formulating appropriate mitigation options.

*Data availability*

Flight measurements are available at the HALO data base at https://halo-db.pa.op.dlr.de/. The SEVIRI data are provided by EUMETSAT (European Organisation for the Exploitation of Meteorological Satellites) and the modelled atmospheric profiles are obtained from ECMWF (European Centre for Medium-Range Weather Forecasts). The GERB data could be accessed from EUMETSAT (GERB international team).

*Author contributions*

Z.W. conducted the analysis and wrote the manuscript. L.B. advised the study and provided feedbacks on the manuscript. L.B., M.W., G.D., S.K., and J.T. plotted Fig. 1, 3, 4, 5a, and 7. C.V. coordinated the ML-CIRRUS mission and L.B. and T. J. contributed to the flight planning. T.J., R.H., H.Z., M.W., S.G., S.Ka., and C.V. participated in the flight measurements. G.D. wrote the description of WALES datasets. Z.W., L.B., T.J., R.H., H.Z., and C.V. discussed the cirrus classification. U.B.

helped with the interpretation of the cirrus cloud conditions and evolution. All authors contributed to and commented on the manuscript.

*Competing interests.* The authors declare that they have no conflict of interest.

*Acknowledgements*

We thank the DLR flight crews for excellent flight operations and EUMETSAT and ECMWF for providing the MSG/SEVIRI observations and modelled atmospheric data, as well as Royal Meteorological Institute of Belgium (RMIB) in the team of the GERB system for supplying GERB-like data. We thank Bernhard Mayer and team for developing the LibRadtran model. This work is supported by the German Research Foundation within SPP-1294 HALO (grant no

VO1504/6-1 and VO1504/7-1) and TRR 301 (Project ID 428312742). Z.W is supported by the DLR (Deutsches Zentrum für Luft- und Raumfahrt) / DAAD (Deutscher Akademischer Austauschdienst) Research Fellowships - Doctoral Studies in Germany, 2020. T.J. also thanks the DLR project H2CONTRAIL. We thank Manuel Gutleben for interesting comments that helped improve the paper.

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
