# Peer review of "Observations of microphysical properties and radiative effects of contrail cirrus and natural cirrus over the North Atlantic"

_Atmospheric Chemistry and Physics, 2022_

## Author Comment (AC1)

Responses to Referee#1 Darrel Baumgardner

We thank all reviewers for their helpful advices and constructive comments about our paper. Their suggestions and criticism have led to a strongly revised and restructured version of our manuscript where we concentrated on two goals: (1) we develop a new method to identify microphysical properties of contrails, contrail cirrus and natural cirrus in the same meteorological conditions from in situ measurements, (2) radiative forcing of contrail cirrus and natural cirrus are derived by satellite observations based radiative transfer modeling in air traffic region favorable for contrails evolution. To this end, we have modified pictures and removed some of them, made the text more concise, added a supplement and wrote clearer explanations.

We thank Darrel Baumgardner for his helpful advices regarding our manuscript acp-2022-537: *Observations of microphysical properties and radiative effects of contrail cirrus and natural cirrus over the North Atlantic*.

In the following we number the referee comments (RC) and give replies (R) to each of them.

**RC1:** This research study, while a worthy topic for investigation, fall far short of its intended goal, i.e. to improve our understanding of the radiative forcing by contrails, contrail cirrus and cirrus by using satellite measurements validated by in situ observations.

**R1:** We thank the reviewer for this comment. We reshaped the focus of our study and now more clearly formulate and address our goals: Based on airborne in-situ and lidar measurements on HALO together with satellite data we identified an ideal contrail cirrus outbreak event in the North Atlantic Region (NAR). With this method we investigated the effective radii that are characteristic of contrails, contrail cirrus and natural cirrus under the given meteorological conditions. We then developed a new method using satellite data and radiative transfer modelling to derive the temporal evolution of the radiative forcing of the contrail cirrus outbreak event and find warming contrail cirrus in the early morning hours and cooling contrail cirrus during the day. We hope that this approach improved the red line of the manuscript. We also changed the title in order to avoid the impression of a systematic study to: *Observations of microphysical properties and radiative effects of a contrail cirrus outbreak over the North Atlantic*

**RC2:** Although the authors have devoted a fair amount of effort to analyze the satellite observations to identify contrails and contrail cirrus, they conclude in the end that it is impossible, so instead they take three passes from in situ measurements and three passes from airborne lidar to conclude, and this is my paraphrasing, "We can't identify contrails or contrail cirrus from the satellite measurements in the region where the aircraft measurements were made, but since we think we might have identified contrails and contrail cirrus with the airborne measurements, we will generalize and assume that there must also be such clouds observed by the satellites". This is not convincing.

**R2:** We didn't state that it is impossible to identify contrails and contrail cirrus in the satellite data and the investigation based on the in situ measurements was not meant to be an alternative to that. Although contrails are visible both in the RGBs and in part in the brightness temperature difference (BTD) plots, it is well known that automated contrail identification in satellite data is a difficult task and only the linear shape of contrails combined with the presence of small ice crystals that induce large BTDs can be used for that task. However, the reviewer's comments show that our arguments were not clear, so we restructured the paper to make its red line much more evident and convey clear conclusions. We agree with the reviewer and removed the discussion on the intercomparison of satellite and in-situ data. Instead we now use the different instruments and methods to identify the dedicated contrail cirrus outbreak event

unambiguously from all available data sources. We also follow the suggestion and produced satellite images with the high spatial resolution, and we clearly find contrail cirrus from satellite, in the same area as the in-situ and lidar observations during the flights as shown in the new figure 2. We derived a new radiative forcing estimate using satellite data and microphysical information from the collocated in situ cirrus measurements in the radiative transfer calculations. We then used this information to derive the RF of the contrail cirrus outbreak as a whole and we investigated the evolution of the contrail cirrus outbreak over a time span of 8 hours from satellite data. This method allows to investigate night and day effects of contrail cirrus, not covered by the airborne data.

**RC3:** I am unconvinced by the arguments that are made by the authors. Whereas case studies are an acceptable means for studying cloud microphysical processes when the data sets are limited and hard to obtain, this study does not fall in that category. There must be thousands of measurements by the DLR Falcon and Halo in contrails and contrail cirrus that could be used and yet the authors have chosen one day with only three passes, with no justification for why this day was chosen.

**R3:** We now give the motivation for the choice of this case study: the cold and humid meteorological situation and the air traffic over the North Atlantic, a region with a relevant impact of air traffic on clouds and radiation (Graf and Schumann, 2012, Duda et al. 2013, Spangenberg et al. 2013, Vázquez-Navarro et al., 2015, Teoh et al. 2022a, b), lead to a contrail cirrus outbreak in a thin cirrus over the open ocean. This day was the "golden day" for contrail cirrus measurements during the ML-CIRRUS campaign. It was predicted three days in advance by the weather and contrail models (see Fig. 4 in Voigt et al., BAMS, 2017), and therefore invited for a case study.

From in situ measurements, the opportunity to identify microphysical properties of contrails, contrail cirrus and natural cirrus in similar meteorological conditions in the Norther Atlantic flight Corridor (NAF) with ordered flight tracks is ideal. Also, with respect to flight operations this flight was unique, in later campaigns regulatory restrictions prevented research flights in the NAF and we were not allowed any more to fly within the NAF perpendicular to the flight tracks of passenger traffic in order to perform in-situ contrail cirrus measurements.

From satellite remote sensing, the condition with contrail cirrus over an open ocean with only a few low-level water clouds and the relative homogeneity of the oceanic background in the solar range and thermal IR increases the ability to retrieve the impact of cirrus on TOA radiation, therefore we selected this day. The computation of contrail cirrus and natural cirrus radiative forcing (Duda et al., 2019) using radiative transfer model calculations in air traffic in high relative humidity regions is a powerful tool (Minnis et al., 2004).

**RC4:** In addition, the lengthy descriptions of the data are overly detailed with unnecessary discussions of irrelevant features. Every sentence has to be written with information that coveys succinctly the point the authors wish the reader to see and understand. There are too much speculations, i.e. "might be", "could be", possibly", with little concrete data that the reader can use to understand what the authors are trying to convey.

**R4:** We thank the referee for this comment, we removed speculations from the manuscript, we shortened the manuscript significantly in order to focus on information which is most relevant for the manuscript. We think that we now more clearly address and convey the messages given by our study.

**RC5:** There are many other aspects of this manuscript that fall short of my expectations, but rather than address them in this review, I will wait for what I hope is a more comprehensive (and convincing) study

that has more in situ measurements in co-located satellite measurements. I will also expect to see a detailed discussion of how the in situ measurements were processed, including an engineering error propagation that includes the expected uncertainties in derived quantities, time offsets in the cloud, NO and RH measurements. quantification of the polarization ratio (not perpendicular to forward but perpendicular to sum of perpendicular and parallel), etc.

**R5:** Our study combines in-situ and remote sensing data to identify a contrail cirrus outbreak situation in the NAF. We then use a new method to derive the RF of contrail cirrus and cirrus based on data from in situ and lidar observations from aircraft and satellite remote sensing used in radiative transfer simulations. We selected this case study on purpose in order to test the proposed methodology. Contrail cirrus in the NAF have been investigated in previous studies (Graf and Schumann., 2012; Duda et al., 2013; Spangenberg et al. 2013, Vázquez-Navarro et al., 2015; Teoh et al. 2022a, b), showing that the NAF is an area of interest and with high variability in contrail cirrus cover and radiative impact.

After testing the methodology in the case study, we will apply it to a more climatological oriented study with a larger dataset to investigate the radiative effects of cirrus and contrail cirrus using RTM simulations involving cloud top height CTH and ice optical thickness IOT from satellite, effect radii $R_{eff}$ from in situ values, and cloud bottom height CBH and cloud thickness from lidar.

In the revised manuscript we give more information on instruments, data evaluation and uncertainties from in-situ and lidar measurements as well as from satellite observations. We give this information in a depth which is required for the manuscript and in a similar detail for all instruments used in this study. We refer to previous publications in the references for deeper insight in instrument issues. In addition, we had to remove results of the polarization ratio from in-situ cloud probes. Here, the reviewer is right, the method and the evaluation of polarization data from the cloud probes needs an in-depth discussion in an independent paper, which is out of the scope of this multi-instrument study.

We have addressed all important points raised by the reviewer in the revised version of the manuscript.

*References*

*Duda, D. P., Minnis, P., Khlopenkov, K., Chee, T. L., and Boeke, R.: Estimation of 2006 Northern Hemisphere contrail coverage using MODIS data, Geophys. Res. Lett., 40, 612–617, doi:10.1002/grl.50097, 2013.*

*Duda, D. P., Bedka, S. T., Minnis, P., Spangenberg, D., Khlopenkov, K., Chee, T., and Smith Jr., W. L.: Northern Hemisphere contrail properties derived from Terra and Aqua MODIS data for 2006 and 2012, Atmos. Chem. Phys., 19, 5313–5330, https://doi.org/10.5194/acp-19-5313-2019, 2019.*

*Graf, K., Schumann, U., Mannstein, H., and Mayer, B.: Aviation induced diurnal North Atlantic cirrus cover cycle, Geophysical Research Letters, 39, https://doi.org/10.1029/2012GL052590, 2012.*

*Minnis, P., Ayers, J., Palikonda, R., and Phan, D. N.: Contrails, Cirrus Trends, and Climate, J. Climate, 17, 1671–16, 2004.*

*Spangenberg, D. A., Minnis, P., Bedka, S. T., Palikonda, R., Duda, D. P., and Rose, F. G.: Contrail radiative forcing over the Northern Hemisphere from 2006 Aqua MODIS data, Geophys. Res. Lett., 40, 595–600, https://doi.org/10.1002/grl.50168, 2013.*

Teoh, R., Schumann, U., Gryspeerdt, E., Shapiro, M., Molloy, J., Koudis, G., Voigt, C., and Stettler, M. E. J.: Aviation contrail climate effects in the North Atlantic from 2016 to 2021, Atmos. Chem. Phys., 22, 10919–10935, https://doi.org/10.5194/acp-22-10919-2022, 2022a.

Teoh, R., Schumann, U., Voigt, C., Schripp, T., Shapiro, M., Engberg, Z., Molloy, J., Koudis, G., and Stettler, M. E. J.: Targeted Use of Sustainable Aviation Fuel to Maximize Climate Benefits, Environmental Science & Technology, 10.1021/acs.est.2c05781, 2022b.

Vázquez-Navarro, M., Mannstein, H., and Kox, S.: Contrail life cycle and properties from 1 year of MSG/SEVIRI rapid-scan images, Atmos. Chem. Phys., 15, 8739–8749, https://doi.org/10.5194/acp-15-8739-2015, 2015.

Voigt, C., Schumann, U., Minikin, A., Abdelmonem, A., Afchine, A., Borrmann, S., Boettcher, M., Buchholz, B., Bugliaro, L., Costa, A., Curtius, J., Dollner, M., Dörnbrack, A., Dreiling, V., Ebert, V., Ehrlich, A., Fix, A., Forster, L., Frank, F., Fütterer, D., Giez, A., Graf, K., Grooß, J.-U., Groß, S., Heimerl, K., Heinold, B., Hüneke, T., Järvinen, E., Jurkat, T., Kaufmann, S., Kenntner, M., Klingebiel, M., Klimach, T., Kohl, R., Krämer, M., Krisna, T. C., Luebke, A., Mayer, B., Mertes, S., Molleker, S., Petzold, A., Pfeilsticker, K., Port, M., Rapp, M., Reutter, P., Rolf, C., Rose, D., Sauer, D., Schäfler, A., Schlage, R., Schnaiter, M., Schneider, J., Spelten, N., Spichtinger, P., Stock, P., Walser, A., Weigel, R., Weinzierl, B., Wendisch, M., Werner, F., Wernli, H., Wirth, M., Zahn, A., Ziereis, H., and Zöger, M.: ML-CIRRUS: The Airborne Experiment on Natural Cirrus and Contrail Cirrus with the High-Altitude Long-Range Research Aircraft HALO, Bulletin of the American Meteorological Society, 98, 271-288, 10.1175/BAMS-D-15-00213.1, 2017.

---

## Author Comment (AC2)

**Responses to anonymous reviewer #2**

We thank all reviewers for their helpful advices and constructive comments about our paper. Their suggestions and criticism have led to a strongly revised and restructured version of our manuscript where we concentrated on two goals: (1) we develop a new method to identify microphysical properties of contrails, contrail cirrus and natural cirrus in the same meteorological conditions from in situ measurements, (2) radiative forcing of contrail cirrus and natural cirrus are derived by satellite observations based radiative transfer modeling in air traffic region favorable for contrails evolution. To this end, we have modified pictures and removed some of them, made the text more concise, added a supplement and wrote clearer explanations.

We thank the referee for highlighting the importance of the study and for helpful comments, which we address in the revision of the manuscript.

In the following we number the referee`s comments (RC) and reply (R) to them individually.

**RC:** Summary of paper:

The authors use a combination of aircraft and satellite measurements, and radiative transfer modeling to analyze a band of thin ice cloud in the North Atlantic air corridor on 26 Mar 2014. (Young) contrails, contrail cirrus, and natural cirrus within the cloud layer are distinguished by in situ measurements of ice particle number concentration and NO gas concentration. The optical thickness, effective radius, and radiative forcing are computed for each cloud type within the cloud band.

General comments:

**RC1**: The goals of the paper are scientifically important and worthy of study, but the authors do not characterize the three cloud types in a convincing manner. The cloud types are defined from in situ measurements, but the authors appear to conflate individual contrail properties to the entire layer at the point of observation. It is not clear what distinguishes a (young) contrail from contrail cirrus, even without the context of the overall cloud band. Several if not most of the contrails appear to be at least two hours old and would likely be visible in the satellite imagery, yet I could find no attempt by the authors to use MSG/SEVIRI satellite observations to classify (or determine the history of) any possible contrail cirrus cloud. In fact, the authors seem to claim that such a distinction is not possible, with an example of an ambiguous contrail encounter between 0843 and 0845 UT to demonstrate the current difficulties in discriminating between young contrails and contrail cirrus. Thus, it seems as though the separation of the cloud observations into different types is essentially meaningless. Add on top of that the difficulties in assigning the properties of individual contrails to the entire cloud layer, the overall usefulness of classifying different points of the cloud as contrail, contrail cirrus, and cirrus is minimal. Although it is clear that new and better definitions of aviation-induced and -influenced ice clouds are necessary, I'm not sure how the authors can proceed to strengthen the paper. Perhaps a more careful study of the numbers and ages of the contrails within a layer may allow for a more useful definition of how much a cirrus layer is influenced by aviation.

**R1a)** author's response

We thank the reviewer for his/her critical discussion of our manuscript which helped to shorten, strengthen and to reshape the manuscript. We now use the aircraft data together with high resolution satellite data to assess the situation and find an ideal contrail cirrus outbreak event, thin natural cirrus

with many contrails and contrail cirrus and hence a large aviation impact. As suggested the study of the numbers and NOx/ages of the contrails and contrail cirrus from airborne in situ or lidar analysis gives information on the magnitude of the aviation influence on the cirrus and helps to achieve consistency between the different data sets with different resolutions.

We note that MSG/SEVIRI satellite observations with the 15min/5min repeat cycle are functions of both time and space and this causes the temporal and spatial difference between in situ and satellite observations. Based on the referee's comments, to ensure a more in-depth understanding, we discuss the evolution of the contrail cirrus outbreak for the entire layer from satellite RGB images, brightness temperature differences (BTDs) and simultaneous air traffic dataset.  We add highly resolved MSG/SEVIRI data which clearly show linear contrails and we consider the evolution of this contrail cirrus outbreak situation in a given area, see new Fig. 2 in the manuscript or Fig. A1 below.  In part, contrails are visible in the BTDs, but not all of them, since their effective radii are not as small as those of fresh contrails and they thus lead to smaller BTDs. In addition, some of the contrails partly overlap in the low resolution BTDs and appear "smeared out". This hinders an automatic detection using e.g. the algorithm by Mannstein et al. (1999, 2010). While we removed the identification of contrails from the satellite images, as suggested by the referee, we overlap in situ information on cirrus on the top of satellite pictures and confirm satellite observed contrail cirrus and natural cirrus with cirrus information from in situ and lidar measurements even considering temporal and spatial difference. We also have added the flight direction. To constrain the manuscript, the analysis about satellite retrieval $R_{eff}$ and IOT based on the in situ cirrus classification was removed.

Moreover, we now focus on the cycle of TOA RF of the contrail cirrus outbreak event in the area of the HALO flight over 8 h from early morning to afternoon and then operate satellite observations based radiative transfer modeling to assess their radiative impact. We find warming contrail cirrus in the early morning and cooling contrail cirrus during the day. We hope that the revised version of the manuscript now more clearly supports the conclusions.

*References*

*Mannstein, H., Meyer, R., and Wendling, P.: Operational Detection of Contrails from NOAA-AVHRR-Data, Int. J. Remote Sens., 20, 1641–1660, 1999.*

*Mannstein, H., Brömser, A., and Bugliaro, L.: Ground-based observations for the validation of contrails and cirrus detection in satellite imagery, Atmos. Meas. Tech., 3, 655–669, https://doi.org/10.5194/amt-3-655-2010, 2010.*

[Figure]

Figure A1: Time series of contrail cirrus and surrounding clouds from MSG/SEVIRI observations over the NAR corridor on 26 March 2014. The first column: RGB-composite with overlaid cirrus, low-level liquid clouds pixels and in situ/lidar leg at close time. The red and green line of the HALO flight track represent contrail cirrus and natural cirrus, respectively. The blue arrow indicates the wind direction, which is almost perpendicular to the line shaped contrail cirrus. The second column: 10.8 μm and 12.0 μm BTD (K) with overlaid cirrus pixels. Blue points show air traffic dataset interpolated to MSG grid from M3 and NATS. The color of the HALO flight track indicates the flight direction. HALO flies from red to blue part. Top to bottom: 08:30, 09:30, 10:00 and 10:30 UTC. The red area is investigated in Sect. 4.2.

Specific comments:

**RC2:** The exposition of the research in the paper is not always easy to follow. For example, it is hard to see the details of the flight path in Figure 2, especially in the blue lines in the lefthand RGB-composite images. The lack of clarity makes it difficult to compare the flight path to the lidar data from Figures 3 and 4. Crucially, the authors never directly inform the reader about the flight path details (including three lidar legs and three in-situ legs) until Figure 6, leading to much confusion for the reader in Section 3.2. Several of the following comments highlight similar difficult-to-follow text.

**R2a)** author's response

Thanks for commenting on this ambiguity. We change Fig. 2 (see comment above) and add highly resolved MSG/Seviri data showing line shaped contrail cirrus during the HALO flight. We replace the entire HALO flight track with the HALO leg close to the time of satellite observation. To make it clear and easy to compare the flight path to the lidar data from Fig. 3 we put Fig. 4 on RHi in the supplementary material S2, and we mark the flight direction of HALO in each plot. We also add information on the wind direction, which is almost perpendicular to the line shaped structures of the contrail cirrus. We modify Fig. 2 and the related text accordingly.

**R2b)** manuscript changes

L229-230: "Figure 2 presents the temporal variation of contrails and surrounding clouds with one HALO in situ/lidar leg at close time and air traffic data 2 to 3 hours before from 08:30 (the first in situ leg) to 10:30 UTC (the third lidar leg)."

L247-253 (caption): "Time series of contrail cirrus and surrounding clouds from MSG/SEVIRI observations over the NAR corridor on 26 March 2014. The first column: RGB-composite with overlaid cirrus, low-level liquid clouds pixels and in situ/lidar HALO leg at close time. The red and green line of the HALO flight track represent contrail cirrus and natural cirrus, respectively. The blue arrow indicates the wind direction almost perpendicular to the line shaped structures of the contrail cirrus. The second column: 10.8 μm and 12.0 μm BTD (K) with overlaid cirrus pixels. Blue points show air traffic dataset interpolated to MSG grid from M3 and NATS. The color of the HALO flight track indicates the flight direction. HALO flies from red to blue part."

**RC3:** Line 235 (Figure 2): It is suggested here, but not entirely clear, but have the blue flight segments in Figure 2 been adjusted to account of the 12-minute difference between the nominal satellite time and the actual time of the image acquisition?

**R3a):** author's response

Thanks for commenting on this ambiguous part. Yes, we have computed the line acquisition time according to the SEVIRI metadata. The cross already indicates the time of simultaneous SEVIRI and HALO measurements. Our new plots contain the direction of flight of HALO (from red to blue). But we removed the comparison of in-situ and remote sensing data, and overlapped satellite images with in situ/lidar leg at close time. Changes in the text are given in the answer section to specific comment #1 concerning the flight track and direction in Fig. 2.

**RC4:** Section 3.1: The peaks in backscatter during Leg 2 look like individual contrails. It is not clear how the lidar observations compare with the HALO aircraft flight path. Figure 2 suggests that most of the flight legs

are perpendicular to the NAR corridor traffic but some legs around 0800 are parallel to NAR corridor traffic. What direction is HALO flying relative to NAR corridor during Legs 1, 2 and 3 in Figure 3?

**R4a):** author's response

Sorry for the unclear description. The HALO flight direction could be determined from the indication of latitude in Fig. 4f, but to show it clearly Figure 2 is now updated with lidar leg 2 at 9:30 and lidar leg 3 at 10:30 and with flight their direction. The HALO flight was perpendicular to the Northern Atlantic Flight (NAF) tracks and perpendicular to the contrail cirrus seen on the satellite images in Fig. 2. In addition, we now give the wind direction almost perpendicular to the line shaped structures of the contrail cirrus. We changed the text accordingly.

**R4b)** manuscript changes

L224-229: "On 26 March 2014 the HALO aircraft started from Oberpfaffenhofen in Germany at approximately 05:30 UTC and probed the cirrus over NAR from around 08:00 to 11:30 UTC with a race track pattern between approx. 51.5°N and 54°N at a longitude of ca. -14°E (-13.6 to -14.4°E), see the flight track in Fig. 1a and also Voigt at al. (2017, Fig.4). In this area, HALO flew 3 lidar legs almost perpendicular to the NAR tracks (07:57 UTC - 08:35 UTC, south to north, 09:17 UTC - 09:30 UTC, north to south, and 10:21 UTC - 10:52 UTC, south to north), each followed by in situ legs at different altitudes."

**RC5:** Line 311: "properties collected during the three legs." Which three legs? The legs described in Figures 3 and 4? Don't the authors state that those are WALES measurement legs and thus "can neither be directly inter-compared nor directly compared to in situ observations taken in between"? How is the reader to know that Figure 5 possible, unless the authors tell the reader beforehand that there are 3 lidar legs and 3 in-situ legs?

**R5a)** author's response

We think that the long sentence here gets the reader confused. Original L310-311 shows that the general overview of $R_{eff}$ against N is obtained from in situ legs. But ultimately the whole sentence was removed to shorten this version of manuscript. In L224 to 229 we have added the general explanation of flight pattern analyzed in our study. In addition, Figure 4 gives detailed information on the flight path, direction and altitude of the lidar and in-situ flight legs.

**RC6:** Figure 6: The reader cannot discern any (young) contrails (blue color) in Figure 6d. I suggest this be removed from the figure. Lines 404 through 406 state that only 1 percent of the observations are (young) contrails.

**R6a)** author's response

We now update Fig. 4d with a new version that gives the cirrus classification from in-situ data more clearly and explicitly. A large fraction of the measured cirrus has been identified as contrail cirrus, in addition some natural cirrus has been measured. Some shot sequences of contrail encounters are also visible, we think this information is helpful to the reader and explains the contrail cirrus outbreak event as observed with in situ data.

RC7: Line 400: The discussion about number concentration (N) at this point appears muddled. "Ncas occurrences decrease by more than 2-3 orders of magnitude from 0.03 to 0.78-0.84 cm^{-3}." Shouldn't this read occurrences increase by more than 2-3 orders of magnitude"?

R7a) author's response

Thanks for commenting on this description. We plot a histogram of used in-situ measured N from CAS and CIP and $\Delta NO$ on 26 March 2014 over NAR. It has a general range of 0 to 1 cm$^{-3}$. We take a value in the middle (0.4 cm$^{-3}$) as separation between high $N_{CAS}$ peaks attributable to contrails and moderate values of $N_{CAS}<0.4$ cm$^{-3}$ that we assign to older contrails/contrail cirrus. The values from 0.03 to 0.78-0.84 cm$^{-3}$ is not the increase of magnitude but the range within which the Ncas occurrences decrease by more than 2-3 orders. The related description is moved to the supplement S2 and changed accordingly. We added Fig. S2 in the supplement to present the histograms of in-situ measured Nice from CAS and CIP and $\Delta NO$ on 26 March 2014 over NAR.

R7b) manuscript changes

Ultimately the sentence was removed in the revised version of manuscript in order not to confuse the reader.

RC8: Section 3.2.1: The discussion in this section implies that most of the contrail cirrus observations are from contrails at least 2 h old. How old are the (young) contrails estimated to be?

R8a) author's response

We thank the referee for your suggestions. The contrails are estimated to > 18 min according to Schumann et al. 1998. Young contrails can only be classified using in situ measurements. In order to avoid confusion, we constrain our new manuscript version to two categories in satellite data, aviation-induced and influenced ice clouds or not from satellite remote sensing. Updates in the Fig. 2 and the text are given in the answer section to general comment #1.

RC9: Lines 407-409: "We finally remark that MSG/SEVIRI satellite observations are left unused for this classification since the distinction between contrail cirrus and natural cirrus from satellite observations is inherently difficult due to the typical characteristics of young contrails - large N - cannot be measured by passive sensors." This statement conflates (young) contrails with contrail cirrus, and would thus make all of the previous discussion from Table 1 classifying each cloud type meaningless.

R9a) author's response

Thanks for your critical comments. Table 1 classify contrails, contrail cirrus and natural cirrus from in situ probed properties. For the aspect of passive satellite observations, the only objective signature of air traffic is the linear shape of the contrails and their higher BTDs since they contain small ice particles. For contrail cirrus there is no objective criterion. We admit the difficulty to distinguish contrails and contrail cirrus, and we now combine contrails and contrail cirrus as the former accounts for a small proportion of the data. Updates in the Fig. 2 are given in the answer section to general comment #1. Ultimately this sentence was removed in the revised version of manuscript.

RC10: Section 3.2.2: This section is poorly worded and misleading. We are not simply looking at the temporal evolution of cloud properties, but variables changing in time and space. The following sentences explain that and thus contradict the beginning sentence. The description of how the SEVIRI measurements are classified according to the HALO observations is a bit unclear. Given that the SEVIRI and HALO measurements might be displaced by as much as 7.5 min, and the total time of the (young) contrail observations is around 110 s (1 percent of 3 h), it it not surprising that the R$_{eff}$ measurements between HALO and CiPS are not correlated, and that no significant difference between the IOT between contrail

snd contrail cirrus was found. Even without the fall streaks, it seems unlikely that the properties of individual (young) contrails can be determined from the SEVIRI data generally.

**R10a**) author's response

We thank the referee for this important comment. We agree and changed the manuscript accordingly. In general, we think that it is possible to investigate the properties of contrails using satellite data, as in Vázquez-Navarro et al. 2015, where contrails were tracked with time. However, the difference in acquisition time between satellite and HALO can lead indeed to mismatches that make it difficult to investigate contrail properties starting from in situ observations. For contrail cirrus the situation is different since these measurements, as shown now in our new plots, are more numerous and form connected regions. Nevertheless, in order to strengthen our paper and reach clear goals we restructured the paper. We have removed Sect 3.2.2 including the analyses about IOT and CTH from satellites, the cirrus classification for MSG/SEVIRI observations according to HALO measurements, and the inter comparison of $R_{eff}$ from in-situ and remote sensing data. Now we use in situ data to characterize microphysical properties of contrails / contrail cirrus / natural cirrus and satellite data to determine the RF of the contrail outbreak.

**RC11**: Lines 453: Most estimates of contrail optical thickness from polar orbiting IR sensors are from clouds at least 2 h old, and thus may not the the young contrails that are implied here. The estimated age of the contrails is not mentioned until line 476 after much discussion about IOT estimates of "contrails". The terms contrail and contrail cirrus are being mixed together and it is unclear what the authors are talking about in this section.

**R11a**) author's response

Thanks for the comments on this section. We took some literature to discuss the variation of IOT as a function of contrail ages. We agree that the term of (young) contrails should not be used here. Ultimately the whole paragraph was removed to shorten this version of manuscript and strengthen the main goals.

**RC12:** Lines 478-479: "From the point of view of optical thickness, the entire cloud seems to be homogeneous without remarkable differences among the cloud types defined in Sect. 3." This statement reinforces the overall lack of utility of the cloud types.

**R12a)** author's response

We still think that the ridge cirrus, at the SEVIRI resolution, has a comparable IOT in the segments that are characterized as contrail cirrus or natural cirrus. Nevertheless, similar with the answer to specific comment #10, we have sharpened our goals, emphasized the difference between in situ and satellite imagery in time and space, and removed Sect 3.2.2 including the analyses about IOT and CTH from satellites, the cirrus classification for MSG/SEVIRI observations according to HALO measurements, and the inter comparison of $R_{eff}$ from in-situ and remote sensing data.

**RC13:** Figure 8: The caption in this figure is not helpful. The cloud properties measured by CiPS are necessarily "at SEVIRI spatial resolution" while the $R_{eff}$ measured by HALO are concurrent and collocated aircraft measurements.

**R13a)** author's response

Sorry, Fig. 8 was removed in the revised version to shorten and strengthen the manuscript.

**RC14:** Line 437 (Figure 8): What times is HALO in the northern part of the race track? Why make the reader determine these times on their own from Figure 6, but not the southern part of the race track?

**R14a)** author's response

We thank the referee for his/her suggestion and make the time of HALO in the northern part of the race track clear in original L437-438 as "In the northern part (i.e. at ~09:15, 10:00, 11:30UTC) of the HALO race track, IOT is between 0.05 and ~0.3, sometimes reaching up to almost 0.4." Ultimately Fig. 8 and the corresponding explanations were removed in this revised version of manuscript.

**RC15:** Lines 528-529: "Finally, we jointly assess radii variations of natural cirrus and adjacent contrail cirrus from CiPS with simultaneous HALO measurements for this NAR case." Please tell the reader that this section refers to Figure 8.

**R15a)** author's response

Thank you for your comment. This section refers to Fig. 9 according to the combination of Fig.8 and the cirrus classification results from original Fig. 6d. We correct text would read "Finally, we jointly assess radii variations of natural cirrus and adjacent contrail cirrus from CiPS with simultaneous HALO measurements for this NAR case based on Fig.8 and the cirrus classification results from Fig. 6d."

We removed the discussion of the comparison of $R_{eff}$ from in situ measurement and satellite remote sensing from the manuscript, as further analysis showed considerable uncertainties.

**RC16:** Lines 535: "The temporal variability of CiPS $R_{eff}$ along the flight path…" The authors again appear to neglect that even satellite measurements are functions of both time and space. Simply say "The variability of CiPS $R_{eff}$ …". Also, say "than that of collocated in situ $R_{eff}$" instead of "than that of simultaneous in situ $R_{eff}$". If the data are averaged over the MSG/SEVERI pixels, they can't be simultaneous. One quantity is time averaged while the other is not.

**R16a)** author's response

Thanks for pointing out this ambiguous part once more. We emphasized that satellite measurements are functions of both time and space and updated "the temporal variability of CiPS $R_{eff}$ along the flight path". Ultimately Fig.8 and the corresponding explanations were removed in this revised version of manuscript.

**RC17:** Section 4.4 and Figure 13: Why include this section? Isn't this redundant because the authors have already compared collocated satellite and HALO observations? Why are the various regional quantities computed until 18 UT when only HALO and SEVIRI observations from 0830 to 1230 UT were presented earlier in the paper? Why does a positive vertical velocity imply "the local downward motion of airmass to warmer temperature layers". Doesn't positive vertical velocity mean upward motion of airmasses?

**R17a)** author's response

Thank you for your points. The reason why this section is included is because "In order to examine microphysical properties and radiative effects of the contrail cirrus outbreak detected in this area and to analyse the corresponding temporal variation", as L429-430 indicates. In the original version of manuscript under discussion, it is also to examine whether cirrus detected during the HALO flight are representative of this area and this time. Thus, this isn't redundant although we have already compared collocated satellite and HALO observations.

We originally computed various regional quantities until 18 UTC to show the diurnal cycle of TOA RF of ice clouds in the area. As HALO left the area around noon, we only present HALO and collocated SEVIRI observations from 0830 to 1130 UTC.

We performed more analyses at early morning around 6 to 7UTC and limited the cycle of TOA RF of the contrail cirrus outbreak in the range of 6 to 14 UTC before and after the HALO flight. The corresponding Fig. A2 in this answer adapted Fig. 7 in the revised version of this manuscript. The new time span helps to between distinguish the positive warming contrail cirrus during the night and early morning and the cooling contrail cirrus outbreak during the day.

The positive or negative vertical velocity are defined in the website of "ERA5 hourly data on pressure levels from 1959 to present" in Climate Data Store. Specifically, "Vertical velocity can be useful to understand the large-scale dynamics of the atmosphere, including areas of upward motion/ascent (negative values) and downward motion/subsidence (positive values)". Thus, the positive vertical velocity means downward motion of airmasses. We removed the vertical velocities from the figure and highlight the most important information, a positive and negative radiative forcing of this contrail cirrus outbreak.

[Figure]

Figure A2: The variations of (a) CC, (b) mean IOT and mean CTH, and (c) SW, LW and net RF within the area indicated by a red box in Fig. 2

R17b) manuscript changes

L454-463: "In Fig. 7a, we observe that CC gradually decreases from 0.77 at 06 UTC to 0.25 at 14 UTC. The positive vertical velocity from ERA5 around that region implies the local downward motion of airmass to warmer temperature layers and the CTH also decreases. IOT in Fig. 7b decreases between 07:00 (0.41) and 10:00 UTC (0.15), then slowly increases until 12 UTC (0.17), then falls to 0.12. CTH decreases during the day and is thus consistent with both the observations of HALO (Fig. 3) and the downward motion. Since an underestimation of CTH by CiPS with respect to WALES (Fig. 3) is observed there, we assume that CTH is also underestimated by CiPS in this area.

Mean net RF over this area in this synoptic situation is positive in the early morning until 9 UTC with the maximum of net RF is at 7 UTC when the sun has risen. Hence the contrail cirrus outbreak is warming during night and early morning hours. After 9 UTC, the forcing becomes negative. More explicitly, from around 9 to 14 UTC the net RF is negative and thus this contrail cirrus outbreak tends to cool during daytime. The strongest cooling is observed at 12 UTC."

Typographic errors and other minor issues:

**RC18:** Lines 162-163: "For CTHs larger than approx. 8 km, CTH has an absolute percentage error of 10%, with underestimation for CTH > 10 km at 50° N and overestimation for CTH < 10 km at the same latitude." What does this mean? That CTH is underestimated when the measured CTH > 10 km but overestimated with the measured CTH < 10 km?

**R18a)** author's response

Yes, the meaning of this sentence is that is.

**RC19:** Line 188: I don't think that "detailly" is a valid word. Perhaps "in detail" would be better here, or simply say that both water and ice clouds are represented in the model (It is assumed that they would be represented realistically as possible by the model.)

**R19a)** author's response

Replaced with "in detail" in L184.

**RC20:** Line 199: Change "transit to" to "transition into".

**R20a)** author's response

Replaced with "transit into" in L200 as it should be a verb.

**RC21:** Figure 2: Time series of contrail cirrus… sounds better than "Temporal variation of contrail cirrus" in the figure title.

**R21a)** author's response

Replaced with "Time series of contrail cirrus" in caption of Fig. 2 in L247.

---

## Author Comment (AC3)

**Responses to anonymous reviewer #3**

We thank all reviewers for their helpful advices and constructive comments about our paper. Their suggestions and criticism have led to a strongly revised and restructured version of our manuscript where we concentrated on two goals: (1) we develop a new method to identify microphysical properties of contrails, contrail cirrus and natural cirrus in the same meteorological conditions from in situ measurements, (2) radiative forcing of contrail cirrus and natural cirrus are derived by satellite observations based radiative transfer modeling in air traffic region favorable for contrails evolution. To this end, we have modified pictures and removed some of them, made the text more concise, added a supplement and wrote clearer explanations.

We thank the reviewer for his/her positive judgement on the manuscript and the helpful comments.

In the following we number the referee`s comments (RC) and reply (R) to them individually.

Summary of paper:

In this paper, the authors investigate an important problem, how to distinguish naturally formed cirrus from contrail cirrus. They use a set of HALO measurements from a flight in the ML-CIRRUS campaign to measure in-situ cirrus properties and gases. This is compared to SEVIRI observations and used combined with a radiative transfer model to estimate the radiative properties of the different cirrus types. This is difficult problem and one of interest to the readers of ACP. The authors have made a good attempt to address this problem, but I would suggest there are some aspects that should be improved before publication.

Main points:

RC1: The results on this work are based on three transects from a single flight. ML-CIRRUS flew through many contrails during the campaign, why is only this set chosen (and could the results/method be easily expanded to other flights?). It is noted that the control NO threshold varies, but is this simple to generalize? I don't think it has to be set manually.

R1a) author's response

We added the reason why only this set is chosen in the abstract, the introduction and Sect. 5 summary and conclusions as suggested by the 1st reviewer, to stress the motivation and significance of this case study. Essentially it was the "golden day" for contrail measurements during the ML-CIRRUS campaign, with predictable contrail conditions in the North Atlantic flight corridor NAF (Voigt et al., 2017, Fig. 4), and a persistent contrail cirrus situation over the North Atlantic region NAR with a blue ocean as background for better sensitivity of the satellite measurements. The contrail coverage and radiative effects with high variability in NAR are important and have been studied in many studies. We took profit of this contrail cirrus outbreak and precious measured data in our paper to derive radiative effects combining in situ measurements and satellite observations. This methodology will be applied to other datasets for future research about the contrail climate impact. Changes in the text are given as follows. The results/method can be expanded to other flights since the background NO thresholds are determined dynamically provided the influence by lightning and wildfires can be excluded. The radiative transfer model simulations that accompany the airborne measurements can also be extended to other suitable campaign flights.

R1b) manuscript changes

L14-18: "On that day, high air traffic density in the NAR combined with large scale cold and humid ambient conditions favoured the formation of a contrail cirrus outbreak situation. In addition, low coverage by low-level water clouds and the homogeneous oceanic albedo increase the sensitivity to retrieve cirrus properties and their radiative effect from satellite remote sensing. This allowed to extend current knowledge on contrail cirrus by combining airborne in situ, lidar and satellite observations."

L469-473: "We choose this contrail cirrus outbreak case because of the large contrail cirrus coverage and high air traffic density. As flight operation in all altitudes is not easily granted due to the high air traffic load in the NAR, the data presented here is also rare and unique in the sense that HALO was able to operate and acquire in-flight measurements of contrail cirrus perpendicular to the flight tracks of the NAR. From satellite remote sensing, few low-level water clouds and the relatively homogeneous oceanic background increase the sensitivity to retrieve cirrus properties."

RC2: A related point, but a lot of the statistics are given in counts, but it is not clear what a count is? Is each one an individual contrail, on SEVIRI pixel, or a second of aircraft time? These will all give different results for the accuracy of any method.

R2a) author's response

A count in the statistics is an aircraft measurement with a frequency of 1 Hz.

R2b) manuscript changes

L331-332: "In total, from 08:30 to 11:30 UTC for each aircraft measurement with a frequency of 1 Hz we have classified 49 contrail observations…"

RC3: The authors spend a considerable amount of time looking at Reff from CIPS. Looking at Strandgren et al (AMT, 2017), it doesn't appear that Reff is validated in that paper. In addition, the comparison to HALO Ref values (Fig. 8c) makes it look like CiPS doesn't have the variability to represent Reff. Does CiPS have the capability (or information) to retrieve Reff?

R3a) author's response

We agree with the reviewer's comment. $R_{eff}$ is calculated according to the concept of $R_{eff} = c[\frac{IWC}{\sigma}]$, where c is 1.64 for ice particles of any shape according to Foot (1988), $\sigma$ is volume extinction coefficient, and $IWC$ is ice water content. This relationship is used for CALIOP (Heymsfield et al., 2005), which is the data with which CiPS was trained, and has been extended to $R_{eff} = 1.64[\frac{IWP}{IOT}]$.. However, the reviewer is right, $R_{eff}$ is not validated in Strandgren et al. 2017a and our first aim was to validate it in this study for contrail cirrus (and the few contrails). Unfortunately, we mixed up results of our evaluation with validation results for $R_{eff}$ from CiPS such that the reader was confused. Thus, to sharpen the red line of our study we decided to remove the $R_{eff}$ analysis from CiPS from the current study.

*Reference*

*Heymsfield, A. J., Winker, D., and van Zadelhoff, G.-J.: Extinction-ice water content-effective radius algorithms for CALIPSO, Geophys. Res. Lett., 32, L10807, 10.1029/2005GL022742, 2005.*

*Foot, J. S.: Some observations of the optical properties of clouds. Part II: Cirrus, Q. J. R. Meteorol. Soc., 114, 145–164, 1988.*

RC4: I am unclear if the extent to which these contrails can or should be considered as temporal evolutions. Fig 8 suggests that they could be a temporal evolution, but around line 300, it is suggested otherwise.

R4a) author's response

Thanks for your consideration. Satellite measurements vary in time and space. To make it clear, we removed Fig 8 and the corresponding description about temporal evolutions of contrails in the text.

RC5: This is more of a style thing, but I found the text could be broken up more (into paragraphs for example) to help the reader. There are several cases were a paragraph spans most of a page (e.g P16), which is too long.

R5a) author's response

Thanks for the suggestions, we adjusted the text accordingly. Around the original L410, and also in all other sections we revised the text and made it more concise and clearer.

Minor points:

RC6: L21 - consistency in the ordering of the cloud types would be nice (perhaps throughout).

R6a) author's response

The order of the cloud types was revised in the throughout text in the sequence of contrails, contrail cirrus and natural cirrus if they exist.

RC7: L163 - This would suggest the CTH is biased towards returning 10km? Does this affect the results?

7a) author's response

Thanks for pointing to this feature. The bias of CTH toward 10 km could lead to a less significant decreasing trend in Fig. 8(a). In Fig. 12, for radiative transfer modeling calculations CTH and CBH from lidar legs are used. These two figures were ultimately removed in the new manuscript version in order to focus and strengthen the manuscript. For Fig. 13, the effects of this bias were explained in original L704-706, "Notice however that the possible underestimation of CTH by CiPS in this area would result in the general underestimation of the LW RF results since a lower CTH reduces the contrast to the cirrus-free OLR. In turn, this would further shift cirrus net RF towards cooling."

RC8: L193 - I would not start a sentence with 'and'. Libradtran recommends this, I assume that is what you used?

R8a) author's response

The "and" has been removed.

RC9: L207 - Presumably this could be checked by looking at the contrail evolution in SEVIRI data

R9a) author's response

Thanks for pointing to this part. To clarify the time when contrails have formed, SEVERI RGBs are produced from 12 UTC of 25 March to 8 UTC of 26 March just before the HALO measurements. We confirmed that contrails identified in this study were induced at 3 UTC on 26 March (see the figures attached below). We removed the last part of the sentence accordingly.

[Figure]

Figure A3: MSG/SEVRI RGB plots at a sequence of time on March 25 and 26 of 2014

R9b) manuscript changes

L209-211: "Considering that the peak of eastbound morning air traffic is approx. at 3 UTC (Graf and Schumann, 2012), under favourable conditions with low temperature and high humidity contrails induced from these aircraft are expected to form and live for hours such that they can be identified in MSG observations in the morning of the same day."

RC10: Fig 1. - This should indicate the study region. It is almost coincident with a MODIS overpass, which could be used for a high resolution check of the contrail properties.

R10a) author's response

We thank for your comments and plotted the flight path in Fig. 1 and adapted the caption to indicate the study region. As for a high resolution check of contrail properties using MODIS images, we have tested before that the MODIS overpass mismatches the HALO flight time between 8 UTC to 11:30 UTC on 26 March. The image at 10:40 and 10:45 UTC show the right edge of NAR. The other three time slots (12:20, 12:30, and 14:10) capture the contrails over the study region but with time difference.

R10b) manuscript changes

L216-217 (Caption): "Figure 1: (a) The false color RGB image from MSG/SEVIRI overlapped with the HALO flight track on 26 March 2014 at 10:45 UTC showing Europe and the Eastern part of the North Atlantic Ocean…"

RC11: L216 - What is the SEVIRI resolution at this location?

R11a) author's response

Approx. 3.5 km × 4.5 km sampling distance

R11b) manuscript changes

L219: "Due to its approx. 3.5 km × 4.5 km spatial resolution…"

RC12: L226 - The first use of NAR?

R12a) author's response

The first use of NAR in the main text is in L91.

RC13: L273 - Do these contrails line up with those observed in SEVIRI? That could give more confidence in the identification.

R13a) author's response

Yes, as suggested by the 2nd reviewer, a visual comparison between measured contrails from HALO instruments and observed ones in SEVIRI at 8:30, 9:30, 10:00, 10:30UTC replaced Fig.2 and a more in-depth analysis and discussion were included in Sect. 3.1, to stress the contrail observation from MSG/SEVIRI. However, as explained there the many lines that can be observed in the high resolution RGBs can be found only partially in the BTDs such that an automatic identification in the BTDs is not possible and a quantitative verification of the contrail locations in in situ and satellite data cannot be achieved.

RC14: L279 - I might have said the ice supersaturation was 'occasional' - the third flight has almost none (if I am reading Fig. 4 correctly).

R14a) author's response

Yes, this is true, especially in leg 3 where supersaturation is limited to small area inside the cloud. We adapted the text with 'occasional'.

RC15: L300 - I would make the temporal comparison (or lack of it) clear earlier (maybe in the flight description).

R15a) author's response

Yes, we updated Fig. 2 and extended the discussion about temporal collocation between HALO measured cirrus and satellite images in Sect. 3.1.

RC16: L306 - aircraft.

R16a) author's response

Updated in the whole main text.

RC17: L327 - Grammar. Also, is this expected? Could it be due to errors in the RH retrieval (or reanalysis)?

R17a) author's response

Thanks for indicating this grammar error. We corrected the sentence accordingly. The uncertainty of the RH retrieval from AIMS measurements is discussed in Sect. 2.1.

R17b) manuscript changes

L135: "…were used to convert water vapor concentration to RHi with an uncertainty of 10 % to 20 % (Kaufmann et al., 2018)."

Original L326-327: "Figure 5b also shows that over the entire flight path R$_{eff}$ increases with RHi, as ice supersaturation supplies the water vapor for the growth of ice crystals, while subsaturated conditions lead**ing** to sublimation and evaporation." But ultimately the whole sentence was removed to shorten this version of manuscript.

*Reference*

*Kaufmann, S., Voigt, C., Heller, R., Jurkat-Witschas, T., Krämer, M., Rolf, C., Zöger, M., Giez, A., Buchholz, B., Ebert, V., Thornberry, T., and Schumann, U.: Intercomparison of midlatitude tropospheric and lower-stratospheric water vapor measurements and comparison to ECMWF humidity data, Atmos. Chem. Phys., 18, 16729-16745, 10.5194/acp-18-16729-2018, 2018.*

RC18: L333 - The previous sentence just noted that different aircraft might produce different NO amounts.

R18a) author's response

Yes, and it explained the reason why the ΔNO threshold could be generalized by using a dynamical NO background value.

RC19: Eq 1 - Using min would also include an impact of instrument noise. Have you thought about using a different measure, perhaps a statistic/algorithm that can remove outliers instead (e.g. RANSAC) for identifying the background?

R19a) author's response

We thank for your comments on a RANSAC algorithm to interpret outliers. We tested your suggested method and confirmed that all outliers correspond to the peaks of NO values, which stress the accuracy of our NO background identification. A supplementary explanation was added in the text.

R19b) manuscript changes

L311-312: "Notably, we use the RANSAC algorithm (Fischler and Bolles, 1981) to interpret NO outliers and confirm that they haven't hit the NO background but the peaks of NO values."

*Reference*

Fischler, M. A and Bolles, R. C.: Random Sample Consensus: A Paradigm for Model Fitting with Applications to Image Analysis and Automated Cartography. Comm. ACM. 24: 381–395. doi:10.1145/358669.358692, 1981.

RC20: Fig. 7 - I like the reduction in aspherical fraction in the contrail region, but is this a consistent effect, or just observed in one case?

R20a) author's response

Yes, it's a consistent effect that aspherical fraction reduces when encountering contrails. But finally, we remove Fig. 7 as it's far away from the main focus of the updated manuscript and because a more in-depths discussion would be needed which is out of the scope of this manuscript.

RC21: Fig. 9 - Given the retrieved Reff has an impact on the optical depth, does the lack of sensitivity to Reff also imply that CiPS is performing poorly when retrieving the IOT? That could potentially explain the difference in optical depths from the expected distribution?

R21a) author's response

The lack of sensitivity to $R_{eff}$ will not influence the IOT retrieval in CiPS. In Sect. 2.2.1, we explained that CiPS consists of four artificial neural networks to detect cirrus with their transparency information and retrieves the corresponding CTH, IOT, and ice water path, respectively. $R_{eff}$ is removed as it's not the direct output of CiPS but the calculations using IWP and IOT.

RC22: L468 - fast -> quickly.

R22a) author's response

Revised but ultimately the whole sentence was removed to shorten this version of manuscript.

RC23: L498 – north. Revised

R23a) author's response

Revised but ultimately the whole sentence was removed to shorten this version of manuscript.

RC24: L598 - derived how?

R24a) author's response

Thanks for pointing out this ambiguous description. $R_{eff}$ profiles are derived using IWC and temperature from ERA5 according to the parameterization by McFarquhar et al. (2003) and Bugliaro et al. (2011, 2022). The equations are listed as follows.

"McFarquhar et al. (2003) is used which relates ice particle effective radius $R_{eff}$ [μm] to ice water content IWC [kg/m3] and temperature T [K]:

$$b = -2.0 + 0.001\sqrt{273 - T}^3 \log((IWC/1000)/(50 g/m^3))$$

$$r_0 = 377.4 + 203.3b + 37.91b^2 + 2.3696b^3$$

$$n_{ft} = (\sqrt{3} + 4)/(3\sqrt{3})$$

$$r_1 = r_0/n_{ft}$$

$$r_{eff} = (4\sqrt{3}/9)r_1$$

R24b) manuscript changes

L394-395: "For liquid clouds, the parameterization by Bugliaro et al. (2011, 2022) are applied for creating $R_{eff}$ profiles using IWC and temperature from ERA5."

*Reference*

*McFarquhar, G., Iacobellis, S., and Somerville, R.: SCM simulations of tropical ice clouds using observationally based parameterizations of microphysics, J. Climate, 16, 1643–1664, 2003.*

RC25: L604 - I was initially skeptical of this, but looking further at CiPS, this doesn't seem so unreasonable. For readers unfamiliar with CiPS, you might want to note that the CiPS retrieval is only dependent on thermal IR channels (which makes it independent of the surface/low cloud properties).

R25a) author's response

Thanks for your kind understanding. I updated the sentence and emphasized that CiPS retrieval is only dependent on thermal channels.

R25b) manuscript changes

L398-399: "Since SEVIRI observations with CiPS are able to account for the entire cirrus cloud layers but are only dependent on thermal channels and not affected by low lying clouds..."

RC26: L618 - What is done for these situations? DO they occur often? Does it impact your results?

R26a) author's response

I see your points and formulated the argumentation. $R_{eff}$ beyond the range of 5 to 60 μm are inexecutable in RTM calculations and not considered in the computations of radiative effects. 20 cases occur in total. It hasn't significant impacts on my results. As presented in updated Fig. 5, $R_{eff}$ of natural cirrus and contrails always fall in the range where RTM could simulate.

R26b) manuscript changes

L414-416: "20 cases in total are removed but have a negligible effect on the estimation of radiative effects as $R_{eff}$ of natural cirrus and contrail cirrus always fall in the range where RTM could simulate as indicated in Fig.5."

RC27: L635 - Is this likely? Perhaps some indication of windspeed at this time would be useful?

R27a) author's response

We thank for your significant advice and re-compute the simulations along the HALO flight track with windspeed from ERA5 as RTM inputs following Cox and Munk (1954a, b) and Nakajima and Tanaka (1983). The Figure A4 in this answer is the updated Fig.6 with the changes of corresponding sentences in the text.

[Figure]

Figure A4: Comparison of TOA (a) RSR and (b) OLR from our RTM simulations (RSR_L, OLR_L) for probed ice particles and RRUMS algorithm results (RSR_R, OLR_R) for single SEVIRI pixel along the HALO flight on 26 March 2014. The mean absolute error (MAE), root mean square error (RMSE) and correlation coefficient (CC) are used as metrics.

R27b) manuscript changes

L396-397: "Besides, the albedo of ocean is parameterized following Cox and Munk (1954a, b) and Nakajima and Tanaka (1983), especially involving the wind speed from ERA5."

L430-432: "Furthermore, a smaller overestimation of RSR by the RTM compared to RRUMS is also observed for the smallest RSR values below 150 W m$^{-2}$, related to the bias of estimated ocean albedo but improved by the application of wind speed."

*Reference*

*Cox, C. and Munk, W.: Measurement of the roughness of the sea surface from photographs of the sun's glitter, J. Opt. Soc. USA, 44, 838–850, 1954a.*

*Cox, C. and Munk, W.: Statistics of the sea surface derived from sun glitter, J. Marine Res., 13, 198–227, 1954b.*

*Nakajima, T. and Tanaka, M.: Effect of wind-generated waves on the transfer of solar radiation in the atmosphere-ocean system, J. Quant. Spectrosc. Radiat. Transfer, 29, 521–537, 1983.*

RC28: L643 - I don't understand this measure of uncertainty or how it is applied here.

R28a) author's response

Thanks for pointing to this incorrect expression. A brief explanation ($RMSE_{RRUMS\_G}$/ mean ($RSR_{RRUMS\_G}$)) was added in the text. But ultimately this measure was removed as it's far away from the main focus of the revised version of the manuscript.

R28b) manuscript changes

Original L643-645: "We consider the ratio of the RMSE value of RSR from RRUMS against GERB ($RMSE_{RRUMS\_G}$/ mean ($RSR_{RRUMS\_G}$)) (Sect. 4.1) divided by the mean RRUMS RSR (ratio=0.19) as a measure for the uncertainty of RRUMS and neglect all RTM simulations that differ by more than this fraction from RRUMS." But ultimately the whole sentence was removed to shorten this version of manuscript.

RC29: Fig. 13c - Is this vertical velocity relevant? Can ERA5 simulate the cirrus vertical velocities at the small scale required for ice processes?

R29a) author's response

ERA5 might miss small scale variability but can give information about larger scale air mass motions that affect for instance relative humidity and temperature. Furthermore, it influences the macrophysical cloud properties for example CTH. Ultimately Fig. 7c was removed in the revised version of manuscript. Adapts to the text is as follows.

R29b) manuscript changes

L454-456: "The positive vertical velocity from ERA5 around that region implies the local downward motion of airmass to warmer temperature layers and the CTH also decreases."

---

## Author Comment (AC4)

[Figure]

Figure 7/A2: The variations of (a) CC, (b) mean IOT and mean CTH, and (c) SW, LW and net RF within the area indicated by a red box in Fig. 2

---

## Referee Report (RR1)

Manuscript Review

Observations of microphysical properties and radiative effects of a contrail cirrus outbreak over the North Atlantic

Ziming Wang et al.

The authors have improved somewhat upon their first effort and partially addressed my previous concerns; however, there remain a number of issues that must be rectified before I can recommend publication. These issues are all related to how the measurements are processed and interpreted from the CAS-POL and CIP, lack of a proper error analysis, and the failure to use the full capabilities of these instruments to distinguish contrail cirrus from cirrus.

1)      The criteria that is used to distinguish contrail cirrus from regular cirrus puzzles me, i.e. it appears that only the relative concentrations of NO and Nice, from the CAS and CIP) are used to discriminate the two types of cirrus. Previous studies, e.g. Järvinen et al. (2016) and Nichman et al (2016) has discussed how the CAS-POL polarization detection is sensitive to the shape of the small ice crystals, and the authors in the present paper also allude to the shape of ice crystals as sensitive to the type of cirrus, and yet neither the polarization ratio from the lidar or the CAS-POL is used to further separate the types of cirrus. Is this because this approach was tried but unsuccessful?

2)      Unless I didn't interpret what was written correctly, it appears that the CAS measurements below 3 μm are not used in the analysis. I assume the thinking is that particles smaller than 3 μm must be aerosol particles, not ice crystals. Whereas that might be a reasonable assumption, from contrail studies that I participated in during the early 1990s, we found that there were significant concentrations of contrail ice crystals smaller than 3 μm (Baumgardner et al., 1998). Similar studies by Kuhn et al. (1996), using the predecessor of the CAS-POL, also documented high concentrations of very small crystals. Then Kleine et al. (2018) also used a CAS-POL over the full size range to detect the smallest ice crystals. Hence, I want to see a reanalysis of the cloud passes using the full range of the CAS-POL since I hypothesize that the difference in contrail cirrus and cirrus will become much more distinct if you are only using number concentration. At the same time, I also hypothesize that the effective radius, Re, will also be much smaller in the contrail cirrus and provide a much more clear separation between regions with contrail cirrus and those without, particularly if you use the particle by particle data to identify fine scale entrainment and mixing.

3)      The use of Kleine et al, (2018) to define the uncertainty in size derivation as ±16% is valid for very small contrail crystals, but not for other crystals. As seen in the figure below, derived from Baumgardner et al., 2016, the uncertainty can be as much as ±50% due to asymmetries in shape. Given that the current study ignores shape as a

parameter in defining cirrus types, this uncertainty is unimportant; however, when comparing Re between contrail and non-contrail cirrus, it becomes important. In addition, the derivation of IWC will be very uncertain when you propagate this uncertainty in the calculation of IWC from the CAS size distribution. The derived IWC will exceed ±100%. Hence, since the N, Re and IWC are incorporated in the radiatice transfer models, these uncertainties will need to be discussed in the model results.

4)      In section 4.1 (not 4.2 as is stated earlier in the manuscript), the radiative model uses an aggregate of ice crystals rather than a more reasonable mix of likely habits. If the average shape was aggregated crystals, where is the evidence from the CIP, which most certainly can identify such aggregates.  Before I am willing to accept this very questionable simplification, I want to see a sensitivity study that show how changing the assumed crystal shapes impacts the resulting radiative fluxes. Likewise, I want to see how the estimated uncertainties in Re and IWC impact the flux calculations.

[Figure]

FIG. 9-3. The percent difference between the maximum dimension of hexagonal ice crystals and the dimension that would be derived from Mie scattering is shown as a function of the maximum dimension for different ARs.

**References**

Baumgardner, D., R.C. Miake-Lye, M.R. Anderson, and R.C. Brown, 1998: An evaluation of temperature, water vapor and vertical velocity structure of an aircraft contrail, J. Geophys. Res., 103,8727-8736.

Baumgardner, D., S. Abel, D. Axisa, R. Cotton, J. Crosier, P. Field, C. Gurganus, A. Heymsfield, A. Korolev, M. Krämer, P. Lawson, G. McFarquhar, J. Z Ulanowski, J. Shik Um, 2016: Chapter 9: Cloud Ice Properties - In Situ Measurement Challenges, AMS Monograph on Ice Formation and Evolution in Clouds and Precipitation: Measurement and Modeling Challenges, Eds. D. Baumgardner, G. McFarquhar, A. Heymsfield, Boston, MA.

Järvinen, E., and Coauthors, 2016: Quasispherical ice in convective clouds. J. Atmos. Sci., 73, 3885–3910, doi:10.1175/JAS-D-15-0365.1.

Kleine, J., Voigt, C., Sauer, D., Schlager, H., Scheibe, M., Jurkat-Witschas, T., et al. (2018). In situ observations of ice particle losses in a young persistent contrail. Geophysical Research Letters, 45, 13,553–13,561

Kuhn, M., A. Petzold, D. Baumgardner, and F.P. Petzold, 1998: Particle composition of a young condensation trail and of upper tropospheric aerosol, Geophys. Res. Lettr., 25, 2679-2682.

Li, Y., Mahnke, C., Rohs, S., Bundke, U., Spelten, N., Dekoutsidis, G., Groß, S., Voigt, C., Schumann, U., Petzold, A., and Krämer, M.: Upper tropospheric slightly ice-subsaturated regions: Frequency of occurrence and statistical evidence for the appearance of contrail cirrus, Atmos. Chem. Phys. Discuss. [preprint], https://doi.org/10.5194/acp-2022-632, in review, 2022.

Nichman, L., Fuchs, C., Järvinen, E., Ignatius, K., Höppel, N. F., Dias, A., Heinritzi, M., Simon, M., Tröstl, J., Wagner, A. C., Wagner, R., Williamson, C., Yan, C., Connolly, P. J., Dorsey, J. R., Duplissy, J., Ehrhart, S., Frege, C., Gordon, H., Hoyle, C. R., Kristensen, T. B., Steiner, G., McPherson Donahue, N., Flagan, R., Gallagher, M. W., Kirkby, J., Möhler, O., Saathoff, H., Schnaiter, M., Stratmann, F., and Tomé, A.: Phase transition observations and discrimination of small cloud particles by light polarization in expansion chamber experiments, Atmos. Chem. Phys., 16, 3651-3664, doi:10.5194/acp-16-3651-2016, 2016.

---

## Author Response (AR2)

of the manuscript ACP-2022-537: "Observations of microphysical properties and radiative effects of a contrail cirrus outbreak over the North Atlantic" by Wang et al.

The authors have improved somewhat upon their first effort and partially addressed my previous concerns; however, there remain a number of issues that must be rectified before I can recommend publication. These issues are all related to how the measurements are processed and interpreted from the CAS-POL and CIP, lack of a proper error analysis, and the failure to use the full capabilities of these instruments to distinguish contrail cirrus from cirrus.

We thank the reviewer Darrel Baumgardner for his helpful advices and constructive comments about our paper. The indicated issues have led to a revised version of our manuscript where we discussed about in situ measurements processing and uncertainty analysis, cirrus classification and the sensitivity on how the assumed crystal shapes, $R_{eff}$ and IWC impact the simulated radiative forcing. To this end, we have written additional explanations and added a table.

In the following we number the referee comments (RC) and give replies (R) to each of them.

**RC1:** The criteria that is used to distinguish contrail cirrus from regular cirrus puzzles me, i.e. it appears that only the relative concentrations of NO and Nice, from the CAS and CIP) are used to discriminate the two types of cirrus. Previous studies, e.g. Järvinen et al. (2016) and Nichman et al (2016) has discussed how the CAS-POL polarization detection is sensitive to the shape of the small ice crystals, and the authors in the present paper also allude to the shape of ice crystals as sensitive to the type of cirrus, and yet neither the polarization ratio from the lidar or the CAS-POL is used to further separate the types of cirrus. Is this because this approach was tried but unsuccessful?

**R1a)** author's response

We thank the reviewer for pointing out this. The assessment of contrails and contrail cirrus particle shapes with cloud probes has been described and studied as part of the Contrail and Cirrus Experiment (CONCERT) in previous publications (Gayet et al., 2012 and Chauvigné et al., 2018). Here mainly young contrails were investigated and a clear trend in microphysical properties with aging was observed. The asymmetry parameter, derived with the polar nephelometer, shows a decrease with aging.

In this study, on one side we do not have measurements of the asymmetry parameter. On the other side, we know that particle shape depends on temperature and humidity as well as on the history of the ice crystals. Here mainly contrails and contrail cirrus of unknown origin (i.e. they could not be traced back to single aircraft) have been detected. The analysis of the shape in a similar way as in Järvinen et al. (2016) would require an enhanced effort, taking into account background atmospheric conditions and the atmospheric conditions along backward trajectories. It is out of the scope of the paper to do this analysis. Further the method to derive the shape for small particles would need a separate paper to discuss everything in detail. This is one of the reasons why we removed the descriptions and plots on asphericity in the first version of this paper.

**RC2:** Unless I didn't interpret what was written correctly, it appears that the CAS measurements below 3 μm are not used in the analysis. I assume the thinking is that particles smaller than 3 μm must be aerosol particles, not ice crystals. Whereas that might be a reasonable assumption, from contrail studies that I participated in during the early 1990s, we found that there were significant concentrations of contrail ice

crystals smaller than 3 μm (Baumgardner et al., 1998). Similar studies by Kuhn et al. (1996), using the predecessor of the CAS-POL, also documented high concentrations of very small crystals. Then Kleine et al. (2018) also used a CAS-POL over the full size range to detect the smallest ice crystals. Hence, I want to see a reanalysis of the cloud passes using the full range of the CAS-POL since I hypothesize that the difference in contrail cirrus and cirrus will become much more distinct if you are only using number concentration. At the same time, I also hypothesize that the effective radius, Re, will also be much smaller in the contrail cirrus and provide a much more clear separation between regions with contrail cirrus and those without, particularly if you use the particle by particle data to identify fine scale entrainment and mixing.

**R2a)** author's response

It is correct that for this study only CAS-DPOL measurements starting at a particle size of 3 μm were used. Measured particles smaller than 3 μm were neglected. The focus of this study is the characterization of a contrail cirrus outbreak – the detection of individual, young contrails (seconds to minutes of age) was not in the scope of the flight strategy for this day due to many reasons. First of all, the operating aircraft HALO is not allowed to fly in the vortex of other aircrafts due to technical reasons. Second, the flight was operated in the North Atlantic flight corridor where HALO had to stay on defined flight paths and could not track individual aircraft pathways.

We agree that ice crystals smaller than 3 μm are most relevant for the microphysical characterization of very young contrails (Voigt et al., 2010; Jeßberger et al, 2013; Kleine et al., 2018) but these young contrails were not detected during this flight.

Kleine et al. (2018) used CAS-DPOL data to study very young contrails with contrail ages smaller than 5 minutes. We removed data below 0.96 μm due to technical issues. These small particles sizes can only be found in very young contrails.

Upon request of the reviewer, we have performed an addition analysis for $R_{eff}$ to address your question through including particles sizes between 0.96 μm and 3 μm. After accounting for smaller particles, the mean $R_{eff}$ for natural cirrus and contrail cirrus decreased by 2%. In contrast, the mean $R_{eff}$ for contrails decreased by 8%. The difference of $R_{eff}$ for contrail cirrus (contrails) and cirrus becomes a little more distinct.

In the manuscript, a short explanation for choosing a lower threshold of 3 μm for the particle size was added. The authors would like to stay consistent with other studies of cirrus and contrail cirrus where a lower limit of the particle size was fixed to 3 μm (Voigt et al., 2017; Righi et al., 2020). Thus, we would like to keep using measurements starting at 3 μm also in this manuscript.

**R2b)** manuscript changes

L307-309: "Since the contrails detected during this flight are older, the lower threshold for the particle size was chosen to be 3 μm to neglect any influence from aerosol particles in the size range below 3 μm. This is consistent with other cirrus and contrail cirrus studies (Voigt at al., 2017; Righi et al., 2020)."

**RC3:** The use of Kleine et al, (2018) to define the uncertainty in size derivation as ±16% is valid for very small contrail crystals, but not for other crystals. As seen in the figure below, derived from Baumgardner et al., 2016, the uncertainty can be as much as ±50% due to asymmetries in shape. Given that the current study ignores shape as a parameter in defining cirrus types, this uncertainty is unimportant; however,

when comparing Re between contrail and non-contrail cirrus, it becomes important. In addition, the derivation of IWC will be very uncertain when you propagate this uncertainty in the calculation of IWC from the CAS size distribution. The derived IWC will exceed ±100%. Hence, since the N, Re and IWC are incorporated in the radiative transfer models, these uncertainties will need to be discussed in the model results.

**R3a)** author's response

We agree with the referee that uncertainties of the size derivation from the probe due to different shapes of the crystals translates into the IWC. This effect is known and we mention it now in the manuscript, thanks for pointing it out. However, $R_{eff}$ rather than N and IWC from in situ data is used as input parameter for the radiative transfer calculations. The IWC for each measurement of $R_{eff}$ corresponding to a vertically homogeneous ice cloud with given IOT from satellite observations is derived using Eq. (6) as the manuscript indicates.

We show below in Table A1 (Table 2 in the revised manuscript) that the $R_{eff}$ and the shape of ice crystals have little effects on the radiative forcing in this case. In both cases, the reason is that the IOT of the pixels is kept constant according to the satellite observations. We finally decided not to update Fig. 7 as the lines from our sensitivity study overlap with the original ones. But we add a new paragraph with the related text at the end of Sect. 4.2 accordingly to address this source of uncertainty.

Table A1: The sensitivity study on how changing the uncertainty of $R_{eff}$ (± 50%) and assumed crystal shapes (aggregates agg and general habit mixture ghm according to Baum et al. (2014)) impact the resulting radiative forcing in Fig. 7.

| UTC/h | Shortwave RF / Wm$^{-2}$) | | | | Longwave RF / Wm$^{-2}$ | | | | Net RF / Wm$^{-2}$ | | | |
|---|---|---|---|---|---|---|---|---|---|---|---|---|
| | agg | -50% | +50% | ghm | agg | -50% | +50% | ghm | agg | -50% | +50% | ghm |
| 6 | | | | | 54.1 | 53.9 | 54.0 | 55.1 | 54.1 | 53.9 | 54.0 | 55.1 |
| 7 | -50.2 | -50.7 | -49.8 | -47.5 | 90.4 | 90.0 | 90.2 | 91.3 | 40.2 | 39.3 | 40.4 | 43.8 |
| 8 | -34.9 | -35.3 | -34.7 | -33.7 | 50.4 | 50.2 | 50.3 | 50.8 | 15.5 | 14.8 | 15.6 | 17.0 |
| 9 | -46.3 | -46.6 | -46.1 | -44.8 | 49.0 | 48.9 | 48.9 | 49.5 | 2.7 | 2.3 | 2.8 | 4.7 |
| 10 | -57.5 | -57.7 | -57.3 | -56.1 | 40.2 | 40.2 | 40.1 | 40.6 | -17.2 | -17.5 | -17.2 | -15.5 |
| 11 | -68.7 | -68.9 | -68.5 | -67.6 | 40.4 | 40.5 | 40.3 | 40.8 | -28.2 | -28.4 | -28.1 | -26.8 |
| 12 | -76.0 | -76.2 | -75.9 | -74.5 | 40.5 | 40.6 | 40.4 | 40.9 | -35.5 | -35.6 | -35.5 | -33.7 |
| 13 | -46.2 | -46.2 | -46.0 | -45.1 | 31.0 | 31.0 | 30.9 | 31.2 | -15.2 | -15.2 | -15.1 | -13.8 |
| 14 | -16.0 | -16.0 | -15.9 | -15.4 | 15.0 | 15.1 | 15.0 | 15.1 | -0.9 | -0.9 | -0.9 | -0.3 |

**R3b)** manuscript changes

L126-128: "The uncertainty of the particle size measurements is ±16% (Kleine et al., 2018) for the lower size range and can be up to ±50% for $R_{eff}$, if the shape of the particles is not known. This translates into an error of up to ±100% for the IWC derived from scattering cloud probes (Baumgardner et al., 2017)".

L454: "… (called $R_{eff,mean}$ in the following), …".

L471-475: "To quantify how the estimated uncertainties in $R_{eff}$ (and resulting IWC) as well as assumed crystal shapes influence the radiative forcing, we compute radiative forcing for aggregates (agg) with $R_{eff}$ = $R_{eff,mean}$ ± 50% and perform a sensitivity study about ice crystal shape using the general habit mixture (ghm) also available from Baum et al. (2014). The simulated values are recorded in Table 2. The uncertainty

of RF due to $R_{eff}$ has an average of 0.2 Wm$^{-2}$ in SW and 0.1 Wm$^{-2}$ in LW. In total the effect on the net RF is approx. 0.1 Wm$^{-2}$…"

**RC4:** In section 4.1 (not 4.2 as is stated earlier in the manuscript), the radiative model uses an aggregate of ice crystals rather than a more reasonable mix of likely habits. If the average shape was aggregated crystals, where is the evidence from the CIP, which most certainly can identify such aggregates. Before I am willing to accept this very questionable simplification, I want to see a sensitivity study that show how changing the assumed crystal shapes impacts the resulting radiative fluxes. Likewise, I want to see how the estimated uncertainties in Re and IWC impact the flux calculations.

**R4a)** author's response

We thank the referee for this comment. We follow the suggestion and perform the sensitivity study on how the assumed crystal shapes impact the resulting radiative forcing. The parameterization named general habit mixture (ghm) from Baum et al (2014) is exploited to represent crystal shapes as a function of crystal size. The ghm with the $R_{eff}$ of 25.2 μm leads to a smaller absolute solar RF and a larger net RF than aggregates. However, the calculated IWC taking the crystal habits as ghm is larger than assuming as aggregates, and then results in a larger solar RF. These two parts cancel each other out in a way and generate a slightly larger net RF from ghm. In general, the impact of assumed crystal shapes on the simulated radiative forcing is small since we keep IOT constant. Additional changes in the text can be found below.

Notably, how the estimated uncertainties in $R_{eff}$ and IWC influence the radiative forcing calculations are given in the answer section to comment RC3.

**R4b)** manuscript changes

L418: "A sensitivity study with respect to ice particle shape is conducted in Sect. 4.2."

L471-476: "To quantify how the estimated uncertainties in $R_{eff}$ (and resulting IWC) as well as assumed crystal shapes influence the radiative forcing, we compute radiative forcing for aggregates (agg) with $R_{eff}$ = $R_{eff,mean}$ ± 50% and perform a sensitivity study about ice crystal shape using the general habit mixture (ghm) also available from Baum et al. (2014).  The simulated values are recorded in Table 2…Compared with aggregates, the ghm model has induced a larger net RF of 1.7 Wm$^{-2}$, with the shift in SW and LW of 1.2 Wm$^{-2}$ and 0.5 Wm$^{-2}$, respectively."

We hope to have addressed the important points raised by the reviewer in the revised version of the manuscript.

*References*

[revised manuscript text omitted]

**Responses to editor**

We thank the editor Farahnaz Khosrawi for her positive judgement on the manuscript and helpful technical corrections.

In the following we number the editor`s comments (EC) and reply (R) to them individually.

Dear authors,

please find enclosed a referee report on the revised version of the manuscript. The referee has still some issues that should be considered/discussed before publication.

Additionally, I would like to ask you to consider the following technical corrections:

EC1: P2, L48-53: This sentence is quite long and difficult to follow. Please consider to shorten or split into two sentences.

R1a) author's response

Yes, we split it into two sentences.

R1b) manuscript changes

L48-53: "Due to various reasons, including the feedback of natural clouds, the radiative response to the presence of contrail cirrus, the uncertainty in upper tropospheric water budget (including initial contrail properties, contrail cirrus properties and relative humidity), contrail cirrus schemes (see Lee et al., 2021), and the challenges in measuring and separating contrail cirrus from natural cirrus, a best central estimate of the contrail cirrus RF remains challenging. It further limits projections of aviation climate impact and formulations of mitigation options other than carbon dioxide (CO2) emissions (Voigt et al., 2021)."

EC2: P3, L108-109: ...and broad and diurnal... -> one "and" too much? Better to use a comma?

R2a) author's response

This sentence means we exploit the information from high resolution (1) airborne measurements, (2) geostationary satellite observations to compute the diurnal cycle of RF in that region. Thus, the first "and" is replaced with "as well as".

R2b) manuscript changes

Updated the "...and broad and diurnal..." with "...as well as geostationary satellite observations with the high repetition rate..."

EC3: P3, L110: section should be written abbreviated as "Sect." except if written at the begin of the sentence, then it is "Section" (see ACP guidelines).

R3a) author's response

Updated all "section" in this paragraph with "Sect."

EC4: P5, L177: to 1% -> to be 1%.

R4a) author's response

Updated the text accordingly.

EC5: P6, Figure 1 caption: Full stop (last sentence) is missing.

R5a) author's response

The full stop "." has been added at the end of this sentence.

EC6: P6, L220: "usually missed" -> please rephrase.

R6a) author's response

Replaced "are usually missed" with "cannot be detected".

EC7: P6, L223: "under study" -> please rephrase.

R7a) author's response

Removed "under study" and replaced "the day" with "26 March 2014".

EC8: P6, L230: add a comma after "before".

R8a) author's response

Updated the text accordingly.

EC9: P6, L234: delete "the" before "Europe".

R9a) author's response

Updated the text accordingly.

EC10: P7, L238: perpendicularly -> perpendicular.

R10a) author's response

Perpendicularly is left here as we think that an adverb is required.

EC11: P9, L259: firstly -> first.

R11a) author's response

Replaced "firstly" with "first".

EC12: P10, L267: can be very probably identified -> can probably be identified

(or write "with high certainty")

R12a) author's response

Replaced "can be very probably identified" with "can probably be identified".

EC13: P11, Figure 4 caption: Add "the" -> over the North Atlantic and put the "N" before "< 3 mu m" and add "and" before "altitude".

R13a) author's response

Updated Figure 4 caption as "In situ measurements of HALO on 26 March 2014 over the North Atlantic region, including (a) ice number concentration N, (b) $R_{eff}$ > 1.5 µm, (c) NO and NO background, (d) cirrus classification, (e) RHi, (f) flight latitude, and (g) altitude".